# Greedy inference with structure-exploiting lazy maps

**Michael C. Brennan** [*]
Massachusetts Institute of Technology
Cambridge, MA 02139 USA
mcbrenn@mit.edu

**Daniele Bigoni** [*]
Massachusetts Institute of Technology
Cambridge, MA 02139 USA
dabi@mit.edu

**Olivier Zahm**
Université Grenoble Alpes, INRIA, CNRS, LJK
38000 Grenoble, France
olivier.zahm@inria.fr

**Alessio Spantini**
Massachusetts Institute of Technology
Cambridge, MA 02139 USA
alessio.spantini@gmail.com

**Youssef Marzouk**
Massachusetts Institute of Technology
Cambridge, MA 02139 USA
ymarz@mit.edu

## Abstract

We propose a framework for solving high-dimensional Bayesian inference problems using *structure-exploiting* low-dimensional transport maps or flows. These maps are confined to a low-dimensional subspace (hence, lazy), and the subspace is identified by minimizing an upper bound on the Kullback–Leibler divergence (hence, structured). Our framework provides a principled way of identifying and exploiting low-dimensional structure in an inference problem. It focuses the expressiveness of a transport map along the directions of most significant discrepancy from the posterior, and can be used to build deep compositions of lazy maps, where low-dimensional projections of the parameters are iteratively transformed to match the posterior. We prove weak convergence of the generated sequence of distributions to the posterior, and we demonstrate the benefits of the framework on challenging inference problems in machine learning and differential equations, using inverse autoregressive flows and polynomial maps as examples of the underlying density estimators.

## 1 Introduction

Inference in the Bayesian setting typically requires the computation of integrals $\int f \, d\pi$ over an *intractable* posterior distribution whose density[2] $\pi$ is known up to a normalizing constant. One approach to this problem is to construct a deterministic nonlinear transformation, i.e., a *transport map* [57], that induces a coupling of $\pi$ with a tractable distribution $\rho$ (e.g., a standard Gaussian). Formally, we seek a map $T$ that pushes forward $\rho$ to $\pi$, written as $T_\sharp \rho = \pi$, such that the change of variables $\int f \, d\pi = \int f \circ T \, d\rho$ makes integration tractable.

Many constructions for such maps have been developed in recent years. Normalizing flows (see [34, 42, 46, 54] and references therein) build transport maps via a deep composition of functions

---

[*]These authors contributed equally to this work.

[2]In this paper, we only consider distributions that are absolutely continuous with respect to the Lebesgue measure on $\mathbb{R}^d$, and thus will use the notation $\pi$ to denote both the distribution and its associated density.

parameterized by neural networks, with certain ansatzes to enable efficient computation. Many recently proposed autoregressive flows (for example [17, 20, 27, 31, 43]) compose triangular maps, which allow for efficient evaluation of Jacobian determinants. In general, triangular maps [9, 33, 47] are sufficiently general to couple any absolutely continuous pair of distributions $(\rho, \pi)$, and their numerical approximations have been investigated in [29, 38, 40, 52]. The flow map of a neural ordinary differential equation [13, 21, 23] can also be seen as an infinite-layer limit of a normalizing flow. Alternatively, Stein variational methods [18, 35, 36] provide a nonparametric way of constructing $T$ as a composition of functions lying in a chosen RKHS.

In general, it can be difficult to represent expressive maps in high dimensions. For example, triangular maps on $\mathbb{R}^d$ must describe $d$-variate functions and thus immediately encounter the curse of dimensionality. Similarly, kernel-based methods lose expressiveness in high dimensions [12, 18]. Flow-based methods often increase expressiveness by adding layers, but this is typically performed in an ad hoc or unstructured way, which also requires tuning.

Here we propose a framework for inference that creates target-informed architectures around *any* class of transport maps or normalizing flows. In particular, our framework uses rigorous *a priori* error bounds to discover and exploit low-dimensional structure in a given target distribution. It also provides a methodology for efficiently solving high-dimensional inference problems via greedily constructed compositions of *structured* low-dimensional maps.

The impact of our approach rests on two observations. First, the coordinate basis in which one expresses a transport map (i.e., $T(x)$ versus $UT(x)$, where $U$ is a rotation on $\mathbb{R}^d$) can strongly affect the training behavior and final performance of the method. Our framework identifies an ordered basis that best reveals a certain low-dimensional structure in the problem. Expressing the transport map in this basis focuses the expressiveness of the underlying transport class and allows for principled dimension reduction. This basis is identified by minimizing an upper bound on the Kullback–Leibler (KL) divergence between $\pi$ and its approximation, which follows from logarithmic Sobolev inequalities (see [59]) relating the KL divergence to gradients of the target density.

Second, in the spirit of normalizing flows, we seek to increase the expressiveness of a transport map using repeated compositions. Rather than specifying the length of the flow before training, we increase the length of the flow sequentially. For each layer, we apply the framework above to a *residual* distribution that captures the deviation between the target distribution and its current approximation. We prove weak convergence of this greedy approach to the target distribution under reasonable assumptions. This sequential framework enables efficient layer-wise training of high-dimensional maps, which especially helps control the curse of dimensionality in certain transport classes. As we shall demonstrate empirically, the greedy composition approach can further improve accuracy at the end of training, compared to baseline methods.

Since Markov chain Monte Carlo (MCMC) methods are also a workhorse of inference, it is useful to contrast them with the variational methods discussed above. In general, these two classes of methods have different computational patterns. In variational inference, one might spend considerable effort to construct the approximate posterior, but afterwards enjoys cheap access to samples and normalized evaluations of the (approximate) target density. How well the approximation matches the true posterior depends on the expressiveness of the approximation class and on one's ability to optimize within this class. MCMC, in contrast, requires continual computational effort (even after tuning), but (asymptotically) generates samples from the exact posterior. Yet there is a line of work that uses transport to improve the performance of MCMC methods ([26, 44])—such that even if one desires exact samples, constructing a transport map can be beneficial. We will demonstrate this link in our numerical experiments.

**Preliminaries.** We will consider target distributions with densities $\pi$ on $\mathbb{R}^d$ that are differentiable almost everywhere and that can be evaluated up to a normalizing constant. Such a target will often be the posterior of a Bayesian inference problem, e.g., $\pi(x) \coloneqq p(x|y) \propto \mathcal{L}_y(x)\pi_0(x)$, where $\mathcal{L}_y(x) \coloneqq p(y|x)$ is the likelihood function and $\pi_0$ is the prior. We denote the standard Gaussian density on $\mathbb{R}^d$ as $\rho$. We will consider maps $T : \mathbb{R}^d \to \mathbb{R}^d$ that are diffeomorphisms,[3] and with some abuse of

notation, we will write the pushforward density of $\rho$ under $T$ as $T_\sharp \rho(x) := \rho \circ T^{-1}(x)|\nabla T^{-1}(x)|$. We will frequently also use the notion of a *pullback* distribution or density, written as $T^\sharp \pi := (T^{-1})_\sharp \pi$.

In §2 we show how to build a single map in the low-dimensional "lazy" format described above, and describe the class of posterior distributions that admit such structure. In §3 we develop a greedy algorithm for building deep compositions of lazy maps, which effectively decomposes any inference problem into a series of lower-dimensional problems. §4 presents numerical experiments highlighting the benefits of the lazy framework. While our numerical experiments employ inverse autoregressive flows [31] and polynomial transport maps [29, 40] as the underlying transport classes, we emphasize that the lazy framework is applicable to any class of transport.

## 2 Lazy maps

Given a unitary matrix $U \in \mathbb{R}^{d \times d}$ and an integer $r \leq d$, let $\mathcal{T}_r(U)$ be the set that contains all the maps $T : \mathbb{R}^d \to \mathbb{R}^d$ of the form

$$T(z) = U \left[ \begin{array}{c} \tau(z_1, \ldots, z_r) \\ z_\perp \end{array} \right] = U_r \tau(z_1, \ldots, z_r) + U_\perp z_\perp \tag{1}$$

for some diffeomorphism $\tau : \mathbb{R}^r \to \mathbb{R}^r$. Here $U_r \in \mathbb{R}^{d \times r}$ and $U_\perp \in \mathbb{R}^{d \times (d-r)}$ are the matrices containing respectively the $r$ first and the $d - r$ last columns of $U$, and $z_\perp = (z_{r+1}, \ldots, z_d)$. Any map $T \in \mathcal{T}_r(U)$ is called a *lazy map* with rank bounded by $r$, as it is nonlinear only with respect to the first $r$ input variables $z_1, \ldots, z_r$ and the nonlinearity is contained in the low-dimensional subspace range$(U_r)$. The next proposition gives a characterization of all the densities $T_\sharp \rho$ when $T \in \mathcal{T}_r(U)$.

**Proposition 1** (Characterization of lazy maps). *Let $U \in \mathbb{R}^{d \times d}$ be a unitary matrix and let $r \leq d$. Then for any lazy map $T \in \mathcal{T}_r(U)$, there exists a strictly positive function $f : \mathbb{R}^r \to \mathbb{R}_{>0}$ such that*

$$T_\sharp \rho(x) = f(U_r^\top x)\rho(x), \tag{2}$$

*for all $x \in \mathbb{R}^d$ where $\rho$ is the density of the standard normal distribution. Conversely, any probability density function of the form $f(U_r^\top x)\rho(x)$ admits a representation as in (2) for some $T \in \mathcal{T}_r(U)$.*

The proof is given in Appendix A.1. By Proposition 1, any posterior density $\pi(x) \propto \mathcal{L}_y(x)\pi_0(x)$ with standard Gaussian prior $\pi_0 = \rho$ and with likelihood function given by $\mathcal{L}_y(x) \propto f(U_r^\top x)$ can be written *exactly* as $\pi = T_\sharp \rho$ for some lazy map $T \in \mathcal{T}_r(U)$. In particular, posteriors of generalized linear models naturally fall into this class; see Appendix D for more details. Following [59, Section 2.1], the solution $T^\star \in \mathcal{T}_r(U)$ to

$$\mathcal{D}_{\mathrm{KL}}(\pi||T_\sharp^\star \rho) = \min_{T \in \mathcal{T}_r(U)} \mathcal{D}_{\mathrm{KL}}(\pi||T_\sharp \rho),$$

is such that $T_\sharp^\star \rho(x) = f^\star(U_r^\top x)\rho(x)$, where $f^\star$ is the conditional expectation

$$f^\star(x_r) = \mathbb{E}\left[ \frac{\pi(X)}{\rho(X)} | U_r^\top X = x_r \right]$$

with $X \sim \rho$.

Now that we know the optimal lazy map in $\mathcal{T}_r(U)$, it remains to find a suitable matrix $U$ and rank $r$. In Appendix A.2 we show that

$$\mathcal{D}_{\mathrm{KL}}(\pi||T_\sharp^\star \rho) = \mathcal{D}_{\mathrm{KL}}(\pi||\rho) - \mathcal{D}_{\mathrm{KL}}((U_r^\top)_\sharp \pi_r||\rho_r), \tag{3}$$

where $\rho_r$ is the density of the standard normal distribution on $\mathbb{R}^r$ and $(U_r^\top)_\sharp \pi$ is the density of $U_r^\top X$ with $X \sim \pi$. Thus, for fixed $r$, minimizing $\mathcal{D}_{\mathrm{KL}}(\pi||T_\sharp^\star \rho)$ over $U$ is the same as finding the most non-Gaussian marginal $(U_r^\top)_\sharp \pi$. Such an optimal $U$ can be difficult to find in practice. The next proposition instead gives a computable *bound* on $\mathcal{D}_{\mathrm{KL}}(\pi||T_\sharp^\star \rho)$, which we will use to construct a $U$ suitable for our algorithm. The proof is given in Appendix A.3.

**Proposition 2.** *Let $(\lambda_i, u_i) \in \mathbb{R}_{\geq 0} \times \mathbb{R}^d$ be the $i$-th eigenpair of the eigenvalue problem $Hu_i = \lambda_i u_i$ where $H = \int (\nabla \log \frac{\pi}{\rho})(\nabla \log \frac{\pi}{\rho})^\top \mathrm{d}\pi$. Let $U = [u_1, \ldots, u_d] \in \mathbb{R}^{d \times d}$ be the matrix containing the eigenvectors of $H$. Then for any $r \leq d$ we have*

$$\mathcal{D}_{\mathrm{KL}}(\pi||T_\sharp^\star \rho) \leq \frac{1}{2}(\lambda_{r+1} + \ldots + \lambda_d). \tag{4}$$

Proposition 2 suggests constructing $U$ as the matrix of eigenvectors of $H$, and that a fast decay in the spectrum of $H$ allows a lazy map with low $r$ to accurately represent the true posterior. Indeed, one can guarantee $\mathcal{D}_{\mathrm{KL}}(\pi||T_\sharp^\star \rho) < \varepsilon$ by choosing $r$ to be the smallest integer such that the left-hand side of (4) is below $\varepsilon$. In practice, since the complexity of representing and training a transport map may strongly depend on $r$, we can bound $r$ by some $r_{\max} \leq d$ associated with a computational budget for constructing $T$. This procedure is summarized in Algorithm 1.

The practical implementation of Algorithm 1 relies on the computation of $H$. Direct Monte Carlo estimation of $H$, however, requires generating samples from $\pi$, which is not feasible in practice. Instead one can use an importance sampling estimate, taking

$$ H \approx \frac{1}{K} \sum_{k=1}^K \omega_k (\nabla \log \frac{\pi}{\rho}(X_k))(\nabla \log \frac{\pi}{\rho}(X_k))^\top, $$

where $\{X_k\}_{k=1}^K$ are i.i.d. samples from $\rho$ and $\omega_k = \frac{\pi(X_k)}{\rho(X_k)}/(\sum_{k'=1}^K \frac{\pi(X_{k'})}{\rho(X_{k'})})$ are self-normalized weights. This estimate can have significant variance when $\rho$ is a poor approximation to the target $\pi$ (e.g., in the first stage of the greedy algorithm in §3). In this case it is preferable to impose $\omega_k = 1$, which reduces variance but yields an biased estimator of $H$; instead, it is an unbiased estimator of $H^{\mathrm{B}} = \int (\nabla \log \frac{\pi}{\rho})(\nabla \log \frac{\pi}{\rho})^\top \mathrm{d}\rho$. As shown via the error bounds in [59, Sec. 3.3.2] this matrix still provides useful information regarding the construction of $U$. We consider the differences between the two estimators in Appendix E. Also, since the effective sample size (ESS) of the importance sampling estimate can be computed with little extra cost after collecting samples, one can use this ESS to choose whether to use $H$ or $H^{\mathrm{B}}$. Other variance reduction methods may also be applicable. For example, simplifications or approximations to the expected outer product of score functions yield natural candidates for control variates.

In constructing a lazy map $T$ of the form (1), one needs to identify a map $\tau : \mathbb{R}^r \to \mathbb{R}^r$ such that $T_\sharp \rho$ approximates the posterior. One can use any transport class to parameterize $\tau$; Appendices B and C detail the particular maps used in our numerical experiments. In our setting we can only evaluate $\pi$ up to a normalizing constant, and thus it is expedient to minimize the reverse KL divergence $\mathcal{D}_{\mathrm{KL}}(T_\sharp \rho||\pi) = \mathcal{D}_{\mathrm{KL}}(\rho||T^\sharp \pi)$, as is typical in variational Bayesian methods—which can be achieved by maximizing a Monte Carlo or quadrature approximation of $\mathbb{E}_\rho \left[ \log T^\sharp \pi \right]$. This is equivalent to maximizing the evidence lower bound (ELBO) and using the reparameterization trick [32] to write the expectation over the base distribution $\rho$. Details on the numerical implementation of Algorithm 1 are given in Appendix F. We note that the lazy framework works to control the KL divergence in the inclusive direction, while optimizing the ELBO minimizes the KL divergence in the exclusive direction. We show empirically that this computational strategy provides performance improvements in both directions of the KL divergence between the true and approximate posterior, compared to a baseline that does not utilize the lazy framework.

---

**Algorithm 1** Construction of a lazy map.

---

1: **procedure** LAZYMAP($\pi, \rho, \varepsilon, r_{\max}$)
2:     Compute $H = \int (\nabla \log \frac{\pi}{\rho})(\nabla \log \frac{\pi}{\rho})^\top \mathrm{d}\pi$
3:     Solve the eigenvalue problem $Hu_i = \lambda_i u_i$
4:     Let $r = r_{\max} \wedge \min\{r \leq d : \frac{1}{2} \sum_{i>r} \lambda_i \leq \varepsilon\}$ and assemble $U = [u_1, \ldots, u_d]$.
5:     Find $T$ by solving $\min_{T \in \mathcal{T}_r(U)} \mathcal{D}_{\mathrm{KL}}(T_\sharp \rho||\pi)$
6:     **return** lazy map $T$
7: **end procedure**

---

## 3 Deeply lazy maps

The restriction $r \leq r_{\max}$ in Algorithm 1 helps control the computational cost of constructing the lazy map, but unless a problem admits sufficient lazy structure, $T_\sharp \rho$ may not adequately approximate the posterior. To extend the numerical benefits of the lazy framework to general problems, we consider the "deeply lazy" map $\mathfrak{T}_\ell$, a composition of $\ell$ lazy maps:

$$ \mathfrak{T}_\ell = T_1 \circ \ldots \circ T_\ell, \quad T_k \in \mathcal{T}_r(U^k), $$

---

**Algorithm 2** Construction of a deeply lazy map

---

1: **procedure** LAYERSOFLAZYMAPS($\pi, \rho, \varepsilon, r, \ell_{\max}$)
2:     Set $\pi_0 = \pi$ and $\ell = 0$
3:     **while** $\ell \leq \ell_{\max}$ and $\frac{1}{2}\operatorname{Tr}(H_\ell) \geq \varepsilon$ **do**
4:         $\ell \leftarrow \ell + 1$
5:         Compute $T_\ell = \text{LAZYMAP}(\pi_{\ell-1}, \rho, 0, r)$            $\triangleright$ Algorithm 1
6:         Update $\mathfrak{T}_\ell = \mathfrak{T}_{\ell-1} \circ T_\ell$
7:         Compute $\pi_\ell = (\mathfrak{T}_\ell)^\sharp \pi$
8:         Compute $H_\ell = \int (\nabla \log \frac{\pi_\ell}{\rho})(\nabla \log \frac{\pi_\ell}{\rho})^\top \mathrm{d}\pi_\ell$
9:     **end while**
10:     **return** $\mathfrak{T}_\ell = T_1 \circ \cdots \circ T_\ell$
11: **end procedure**

---

where each $T_k$ is a lazy map associated with a different unitary matrix $U^k \in \mathbb{R}^{d \times d}$. For simplicity we consider the case where each lazy layer $T_k$ has the same rank $r$, though it is trivial to allow the ranks to vary from layer to layer. In general, the composition of lazy maps is not itself a lazy map. For example, there exists $U^1 \neq U^2$ such that $\mathfrak{T}_2 = T_1 \circ T_2$ can depend nonlinearly on each input variable and so $\mathfrak{T}_2$ cannot be written as in (1).

The diagnostic matrix $H$ allows us to build deeply lazy maps in a greedy way. After $\ell - 1$ iterations, the composition of maps $\mathfrak{T}_{\ell-1} = T_1 \circ \ldots \circ T_{\ell-1}$ has been constructed. We seek a unitary matrix $U^\ell \in \mathbb{R}^{d \times d}$ and a lazy map $T_\ell \in \mathcal{T}_r(U^\ell)$ such that $(\mathfrak{T}_{\ell-1} \circ T_\ell)_\sharp \rho$ best improves over $(\mathfrak{T}_{\ell-1})_\sharp \rho$ as an approximation to the posterior. To this end, we define the residual distribution

$$\pi_{\ell-1} = (\mathfrak{T}_{\ell-1})^\sharp \pi,$$

i.e., the pullback of $\pi$ through the current transport map $\mathfrak{T}_{\ell-1}$. Note that $\mathcal{D}_{\text{KL}}(\pi || (\mathfrak{T}_{\ell-1} \circ T_\ell)_\sharp \rho) = \mathcal{D}_{\text{KL}}(\pi_{\ell-1} || (T_\ell)_\sharp \rho)$. We thus build $T_\ell$ using Algorithm 1, replacing the posterior $\pi$ by the residual distribution $\pi_{\ell-1}$. We then update the transport map to be $\mathfrak{T}_\ell = \mathfrak{T}_{\ell-1} \circ T_\ell$ and the residual density $\pi_\ell = (\mathfrak{T}_\ell)^\sharp \pi$.

We note that applying Proposition 2 to $\pi_\ell$ with $r = 0$ yields

$$\mathcal{D}_{\text{KL}}(\pi || (\mathfrak{T}_\ell)_\sharp \rho) = \mathcal{D}_{\text{KL}}(\pi_\ell || \rho) \leq \frac{1}{2}(\lambda_1 + \cdots + \lambda_d) = \frac{1}{2}\operatorname{Tr}(H_\ell),$$

where we define the diagnostic matrix at iteration $\ell$ as,

$$H_\ell = \int \left( \nabla \log \frac{\pi_\ell}{\rho} \right) \left( \nabla \log \frac{\pi_\ell}{\rho} \right)^\top \mathrm{d}\pi_\ell.$$

Our framework thus naturally exposes the error bound $\frac{1}{2}\operatorname{Tr}(H_\ell)$ on the forward KL divergence, which is of independent interest and applicable to *any* flow-based method. We refer to this bound as the *trace diagnostic*.

This bound can also be used as a stopping criterion for the greedy algorithm; one can continue adding layers until the bound falls below some desired threshold. This construction is summarized in Algorithm 2, and details on its numerical implementation are given in Appendix F.

The next proposition gives a sufficient condition on $U^\ell$ to guarantee the convergence of our greedy algorithm. The proof is given in Appendix A.4.

**Proposition 3.** *Let $U^1, U^2, \ldots$ be a sequence of unitary matrices. For any $\ell \geq 1$, we let $T_\ell \in \mathcal{T}_r(U^\ell)$ be a lazy map that minimizes $\mathcal{D}_{\text{KL}}(\pi_{\ell-1} || (T_\ell)_\sharp \rho)$, where $\pi_{\ell-1} = (T_1 \circ \ldots \circ T_{\ell-1})^\sharp \pi$. If there exists $0 < t \leq 1$ such that for any $\ell \geq 1$*

$$\mathcal{D}_{\text{KL}}((U_r^{\ell\top})_\sharp \pi_{\ell-1} || \rho_r) \geq t \sup_{\substack{U \in \mathbb{R}^{d \times d} \\ s.t. \ UU^\top = I_d}} \mathcal{D}_{\text{KL}}((U_r^\top)_\sharp \pi_{\ell-1} || \rho_r), \tag{5}$$

*then $(T_1 \circ \ldots \circ T_\ell)_\sharp \rho$ converges weakly to $\pi$.*

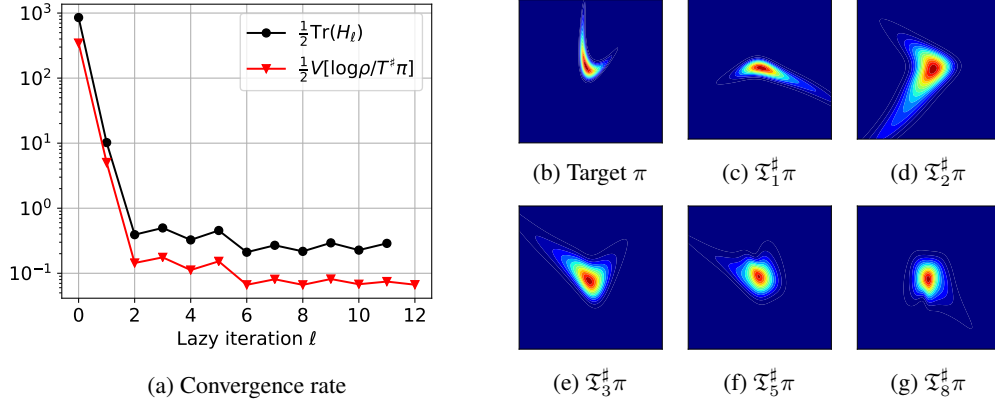

(a) Convergence rate     (b) Target $\pi$    (c) $\mathfrak{T}_1^\sharp \pi$    (d) $\mathfrak{T}_2^\sharp \pi$    (e) $\mathfrak{T}_3^\sharp \pi$    (f) $\mathfrak{T}_5^\sharp \pi$    (g) $\mathfrak{T}_8^\sharp \pi$

Figure 1: Convergence of the algorithm for the approximation of the rotated banana distribution. (a) Decay of the bound $\frac{1}{2}\operatorname{Tr}(H_\ell^{\mathrm{B}})$ on the KL-divergence $\mathcal{D}_{\mathrm{KL}}(\pi\|(\mathfrak{T}_\ell)_\sharp\rho)$ and the variance diagnostic $\frac{1}{2}\mathbb{V}_\rho[\log\rho/\mathfrak{T}_\ell^\sharp\pi]$. (b) The target density $\pi$. (c–g) The target distribution is progressively Gaussianized by the maps $\mathfrak{T}_\ell$.

Let us comment on the condition (5). Recall that the unitary matrix $U$ that maximizes $\mathcal{D}_{\mathrm{KL}}((U_r^\top)_\sharp\pi_{\ell-1}\|\rho_r)$ is optimal; see (3). By (5), the case $t=1$ means that $U^\ell$ is optimal at each iteration. This corresponds to an *ideal* greedy algorithm. The case $0 < t < 1$ allows suboptimal choices for $U^\ell$ without losing the convergence property of the algorithm. Such a greedy algorithm converges even with a potentially crude selection of $U^\ell$ that corresponds to a $t$ close to zero. This also is why an approximation to $H_\ell$ is expected to be sufficient; see Section 4. We emphasize that condition (5) must apply simultaneously to *all* layers for a given $0 < t \le 1$. Following [55], one could relax this condition by replacing $t$ with a sequence $(t_\ell)$ that goes to zero sufficiently slowly. This development is left for future work. Finally, note that Proposition 3 does not require any constraints on $r$, so we have convergence even with $r=1$, where each layer only acts on a single direction at a time.

## 4   Numerical examples

We present numerical demonstrations of the lazy framework as follows. We first illustrate Algorithm 2 on a 2-dimensional toy example, where we show the progressive Gaussianization of the posterior using a sequence of 1-dimensional lazy maps. We then demonstrate the benefits of the lazy framework (Algorithms 1 and 2) in several challenging inference problems. We consider Bayesian logistic regression and a Bayesian neural network, and compare the performance of a baseline transport map to lazy maps using the same underlying transport class. We measure performance improvements in four ways: (1) the final ELBO achieved by the transport maps after training; (2 and 3): the final trace diagnostics $\frac{1}{2}\operatorname{Tr}(H_\ell^{\mathrm{B}})$ and $\frac{1}{2}\operatorname{Tr}(H_\ell)$, which bound the error $\mathcal{D}_{\mathrm{KL}}(\pi\|(\mathfrak{T}_\ell)_\sharp\rho)$; and (4) the *variance diagnostic* $\frac{1}{2}\mathbb{V}_\rho[\log\rho/\mathfrak{T}_\ell^\sharp\pi]$, which is an asymptotic approximation of $\mathcal{D}_{\mathrm{KL}}((\mathfrak{T}_\ell)_\sharp\rho\|\pi)$ as $(\mathfrak{T}_\ell)_\sharp\rho \to \pi$ (see [40]). Finally, we highlight the advantages of greedily training lazy maps in a nonlinear problem defined by a high-dimensional elliptic partial differential equation (PDE), often used for testing high-dimensional inference methods [4, 16, 53]. Here, the lazy framework is needed to make variational inference tractable by controlling the total number of map parameters. We also illustrate the utility of such flows in preconditioning Markov chain Monte Carlo (MCMC) samplers [26, 44], or equivalently as a way of de-biasing the variational approximation on these three problems.

Numerical examples are implemented [4] both in the `TransportMaps` framework [7] and using the TensorFlow probability library [19]. The PDE considered in 4.4 is discretized and solved using the `FEniCS` [37] and `dolfin-adjoint` [22] packages.

## 4.1 Illustrative toy example

We first apply the algorithm on the standard problem of approximating the rotated banana distribution $Q_\sharp \pi_{X_1, X_2}$ defined by $X_1 \sim \mathcal{N}(0.5, 0.8)$ and $X_2 | X_1 \sim \mathcal{N}(X_1^2, 0.2)$, and where $Q$ is a random rotation. We restrict ourselves to using a composition of rank-1 lazy maps. We consider degree 3 polynomial maps as the underlying transport class. We use Gauss quadrature rules of order 10 for the discretization of the KL divergence and the approximation of $H_\ell^B$ ($m = 121$ in Algorithm 3 and 5). Figure 1b shows the target distribution $\pi := \pi_{X_1, X_2}$. Figure 1a shows the convergence of the algorithm both in terms of the trace diagnostic $\frac{1}{2} \operatorname{Tr}(H_\ell^B)$ and in terms of the variance diagnostic. After two iterations the algorithm has explored all directions of $\mathbb{R}^2$, leading to a fast improvement. The convergence stagnates once the trade-off between the complexity of the underlying transport class and the accuracy of the quadrature has been saturated. Figures 1c–g show the progressive Gaussianization of the residual distributions $\mathfrak{T}_\ell^\sharp \pi$ for different iterations $\ell$.

## 4.2 Bayesian logistic regression

We now consider a high-dimensional Bayesian logistic regression problem using the UCI Parkinson's disease classification data [1], studied in [49]. We consider the first 500 provided attributes consisting mainly of patient audio extensions. This results in a $d = 500$ dimensional inference problem. We choose a relatively uninformative prior of $\mathcal{N}(0, 10^2 I_d)$. Here we consider inverse autoregressive flows (IAFs) [31] for the underlying transport class. Details on the IAF structure, our choice of hyper-parameters, and training procedure are in Appendix C.

As noted in §2 and shown in Appendix D, generalized linear models can admit an exactly lazy structure, where the lazy rank $r$ of the posterior is bounded by the number of observations. We demonstrate this by first considering a small subset of 20 observations. Given a sufficiently expressive underlying transport class, a single lazy map of rank $r = 20$ can exactly capture the posterior. We compare four transport maps: a baseline IAF map; $U$-IAF, which is a 1-layer lazy map with rank $r = d = 500$ expressed in the computed basis $U$; $U_r$-IAF and $U_r$-IAF-500, which are 1-layer lazy maps of rank $r = 20$. The baseline IAF, $U$-IAF and $U_r$-IAF each use autoregressive networks with a hidden dimension equal to the input dimension of the flow ($d = 500$ in the case of the baseline IAF and $U$-IAF, 20 in the case of $U_r$-IAF). For $U_r$-IAF-500, we use a hidden dimension of 500, resulting in a map with approximately the same number of flow parameters as the baseline and $U$-IAF maps. Results are summarized in Table 1. We see improved performance in each of the lazy maps compared to the baseline. We also note that $U_r$-IAF outperforms $U$-IAF in each metric median, even though the $U$-IAF map has more flow parameters than the $U_r$-IAF map (4008000 vs 6720). The $U_r$-IAF-500 map performs the best in each metric. This map has the highest ratio of map parameters to active dimensions. This highlights a key benefit of the lazy framework: the ability to focus the expressiveness of a transport map along particular subspaces important to the capturing the posterior.

Next we consider a full rank Bayesian logistic regression problem using 605 observations. Here we compare a baseline IAF; $U$-IAF defined as before; and a 3-layer lazy map trained via the greedy Algorithm 2, denoted G3-IAF. In G3-IAF, each layer has rank $r = 200$. Results are summarized in Table 1, and again we see improvements in each of the performance metrics compared to the baseline IAF. Recall that the basis $U$ relates to a bound on the inclusive KL direction, while the objective function for map training within a layer optimizes the exclusive KL direction. Empirically we see benefits in metrics relating to both directions. Interestingly, we observe that $U$-IAF achieves the greatest ELBO while G3-IAF achieves the lowest trace diagnostics. This suggests that using a larger number of lazy layers tends to lead to improvements to the inclusive KL divergence. Also, though we chose to use the same number of training iterations in each case, we observe that training of the lazy maps converges more quickly; see Appendix G.1 for addition details.

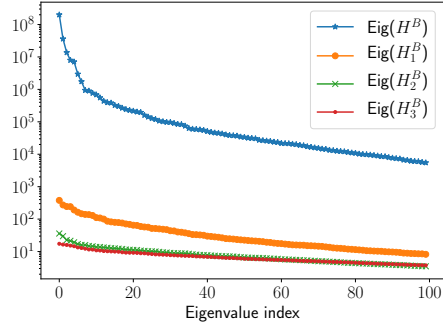

Figure 2: Leading eigenvalues of the diagnostic matrices $H_\ell^B$ for the G3-IAF map applied to the full rank logistic regression problem. The spectrum flattens and falls as the approximation to the posterior improves.

Table 1: Result summaries for the Bayesian logistic regression and Bayesian neural network examples. Values reported are the median and (interquartile range) across 10 trials with randomized initialization. Best performance is bolded. $\Delta$ ELBO computed using the median of the baseline.

| Map | $\Delta$ ELBO* ($\uparrow$) | Variance diagnostic ($\downarrow$) | $\text{Tr}(H_\ell^{\text{B}})/2$ ($\downarrow$) | $\text{Tr}(H_\ell)/2$ ($\downarrow$) |
|---|---|---|---|---|
| *Low rank Bayesian logistic regression* | | | | |
| Baseline IAF | – | 26.5 (3.88) | 104 (9.19) | 31.3 (15.6) |
| $U$-IAF | 6.72 (0.469) | 7.48 (1.37) | 37.4 (2.83) | 20.2 (7.91) |
| $U_r$-IAF | 8.27 (0.249) | 5.68 (0.935) | 33.8 (4.29) | 19.1 (7.68) |
| $U_r$-IAF-500 | **10.9** (0.227) | **1.66** (0.496) | **8.89** (6.63) | **6.19** (0.896) |
| *Full rank Bayesian logistic regression* | | | | |
| Baseline IAF | – | 209 (21.4) | 956 (68.6) | 350 (178) |
| $U$-IAF | **26.4** (1.19) | 130 (12.3) | 623 (19.5) | 287 (96.1) |
| G3-IAF | 1.68 (1.56) | **109** (8.25) | **510** (21.6) | 219 (110) |
| *Bayesian neural network* | | | | |
| Baseline Affine | – | 1.6e4 (5.8e4) | 3.5e5 (6.9e5) | 960 (1.0e3) |
| G3-Affine | **47.7** (2.33) | **97.5** (6.47) | **1.06e3** (56.2) | **606** (201) |

As discussed in the introduction, a powerful use case for transport maps is the ability to precondition an MCMC method as described in [26, 44, 45], i.e., using the computed map to improve the posterior geometry. Applying Hamiltonian Monte Carlo [41] to the full rank Bayesian logistic regression problem (in particular, sampling the pullback $\mathfrak{T}_\ell^\sharp \pi$ where $\mathfrak{T}_\ell$ is the learned $U$-IAF map), we achieve worst, best, and average component-wise effective sample sizes of 0.39%, 1.8%, and 0.99%, compared to 0.056%, 0.12%, and 0.065% without a transport map (sampling the target $\pi$ directly). Note that applying $\mathfrak{T}_\ell$ to MCMC samples from the pullback yields asymptotically exact samples from $\pi$. Three leapfrog steps were used in the HMC proposal, and the step sizes were chosen adaptively during the burn-in period of the chains to obtain acceptance rates between 70% and 90% [3, 5, 6].

## 4.3 Bayesian neural network

We now consider a Bayesian neural network, also in [18, 36], trained on the UCI yacht hydrodynamics data set [2]. Our inference problem is 581-dimensional, given a network input dimension of 6, one hidden layer of dimension 20, and an output layer of dimension 1. We use sigmoid activations in the input and hidden layer, and a linear output layer. Model parameters are endowed with independent Gaussian priors with zero mean and variance 100. Further details are in Appendix G.2.

Here we consider affine maps as the underlying class of transport. This yields Gaussian approximations to the posterior distribution in both the lazy and baseline cases. We compare a baseline affine map and G3-affine, denoting a 3-layer lazy map where each layer has rank $r = 200$. The diagnostic matrices $H_\ell^{\text{B}}$ are computed using 581 standard normal samples. We note improvements in each of the performance metrics using the lazy framework, summarized in Table 1. We also note a 64% decrease in the number of trained flow parameters in G3-affine, relative to the baseline case (from 338142 to 120600).

Similarly to §4.2, we compare the performance of HMC applied with and without transport map preconditioning. We achieve worst, best, and average component-wise ESS of 0.073%, 1.2%, and 0.56% using the learned $G3$-Affine map, compared to 0.047%, 0.14%, and 0.06% without a transport map. Here five leapfrog steps were used in the HMC proposal, and the step sizes in each case were picked adaptively as before.

## 4.4 High-dimensional elliptic PDE inverse problem

We consider the problem of estimating the diffusion coefficient $e^{\kappa(\boldsymbol{x})}$ of an elliptic PDE from sparse observations of the field $u(\boldsymbol{x})$ solving

$$\begin{cases} \nabla \cdot (e^{\kappa(\boldsymbol{x})} \nabla u(\boldsymbol{x})) = 0, & \text{for } \boldsymbol{x} \in \mathcal{D} := [0,1]^2 , \\ u(\boldsymbol{x}) = 0 \text{ for } \boldsymbol{x}_1 = 0, \ u(\boldsymbol{x}) = 1 \text{ for } \boldsymbol{x}_1 = 1, \ \frac{\partial u(\boldsymbol{x})}{\partial \boldsymbol{n}} = 0 \text{ for } \boldsymbol{x}_2 \in \{0,1\} . \end{cases} \tag{6}$$

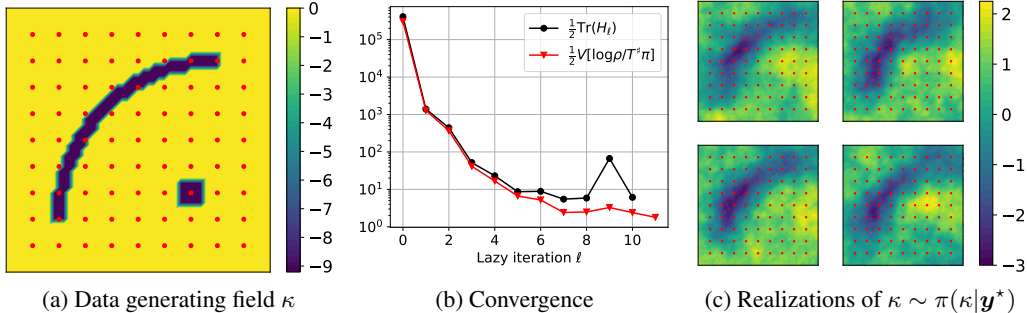

| (a) Data generating field $\kappa$ | (b) Convergence | (c) Realizations of $\kappa \sim \pi(\kappa|\boldsymbol{y}^\star)$ |

Figure 3: Application of Algorithm 2 to an elliptic PDE with unknown diffusion coefficient. (a) The data-generating field $\kappa$. (b) Convergence of the trace error bound and variance diagnostic with greedy iterations. (c) Draws from the 2601-dimensional posterior distribution.

This PDE is discretized using finite elements over a uniform mesh of $51 \times 51$ nodes, leading to $d = 2601$ degrees of freedom. We denote by $\boldsymbol{\kappa}$ the discretized version of the log-diffusion coefficient over this mesh. Let $\mathcal{F}$ be the map from the parameter $\boldsymbol{\kappa}$ to $n = 81$ values of $u$ collected at the locations shown in Figure 3a. Observations follow the model $\boldsymbol{y} = \mathcal{F}(\boldsymbol{\kappa}) + \epsilon$, where $\epsilon \sim \mathcal{N}(0, \Sigma_{\text{obs}})$ and $\Sigma_{\text{obs}} \coloneqq 10^{-3} I_d$. The coefficient $\boldsymbol{\kappa}$ is endowed with a Gaussian prior $\mathcal{N}(0, \Sigma)$ where $\Sigma$ is the covariance of an Ornstein–Uhlenbeck process. For the observations $\boldsymbol{y}^\star$ associated to the parameter $\boldsymbol{\kappa}^\star$ shown in Figure 3a, our target distribution is $\pi(\boldsymbol{z}) \propto \mathcal{L}_{\boldsymbol{y}^\star}(\boldsymbol{z})\rho(\boldsymbol{z})$, where $\boldsymbol{\kappa} = \Sigma^{1/2}\boldsymbol{z}$.

We greedily train a deeply lazy map using Algorithm 2, using triangular polynomial maps as the underlying transport (see Appendix B). Expectations appearing in the algorithm are discretized with $m = 500$ Monte Carlo samples. To not waste work in the early iterations, we use affine maps of rank $r = 4$ for iterations $\ell = 1, \ldots, 5$. Then we switch to polynomial maps of degree 2 and rank $r = 2$ for the remaining iterations. This reflects the flexibility of the lazy framework; changes to the underlying transport class and the lazy rank of each layer are simple to implement. The algorithm is terminated when it stagnates after exhausting the expressiveness of the underlying transport class, and the precision of approximating the objective using $m$ samples; see Figure 3b. Randomly drawn realizations of $\boldsymbol{\kappa}$ in Figure 3c resemble the generating field.

This elliptic PDE is a challenging benchmark problem for high-dimensional inference [4, 16, 53]. We note that the final map is a *sparse* degree-32 polynomial that acts nonlinearly on all 2061 degrees of freedom. Without imposing structure, the curse of dimensionality would render the solution of this problem using polynomial transport maps completely intractable [56]. For instance, a naïve total-degree parameterization of just the final component of the map would contain $\binom{2061+32}{32} \approx 5.5 \times 10^{70}$ parameters. We can confirm the quality of the posterior approximation and demonstrate a further application of transport by using MCMC to sample the pullback $\mathfrak{T}_\ell^\sharp \pi$. We do so using *preconditioned Crank-Nicolson* (pCN) MCMC [15] (a state-of-the-art algorithm for PDE problems, with dimension-independent convergence rate) with a step size parameter $\beta = 0.5$. The acceptance rate is $28.2\%$ with the worst, best, and average effective sample sizes [58] being $0.2\%$, $2.6\%$, and $1.5\%$ of the complete chain. For comparison, a direct application of pCN with the same $\beta$ leads to an acceptance rate under $0.4\%$ and an effective sample size that cannot be reliably computed. More details are in Appendix G.3.

## 5  Conclusions

We have presented a framework for creating target-informed architectures for transport-based variational inference. Our approach uses a rigorous error bound to identify low-dimensional structure in the target distribution and focus the expressiveness of the transport map or flow on an important subspace. We also introduce and analyze a greedy algorithm for building deep compositions of low-dimensional maps that can iteratively approximate general high-dimensional target distributions. Empirically, these methods improve the accuracy of inference, accelerate training, and control the complexity of flows to improve tractability. Ongoing work will consider constructive tests for further varieties of underlying structure in inference problems, and their implications on the structure of flows.

## Broader Impact

**Who may benefit from this research?**    We believe users and developers of approximate inference methods will benefit from our work. Our framework works as an "outer wrapper" that can improve the effectiveness of any flow-based variational inference method by guiding its structure. We hope to make expressive flow-based variational inference more tractable, efficient, and broadly applicable, particularly in high dimensions, by developing automated tests for low-dimensional structure and flexible ways to exploit it. The trace diagnostic developed in our work rigorously assesses the quality of transport/flow-based inference, and may be of independent interest.

**Who may be put at disadvantage from this research?**    We don't believe anyone is put at disadvantage due to this research.

**What are the consequences of failure of the system?**    We specifically point out that one contribution of this work is identifying when a poor posterior approximation has occurred. A potential failure mode of our framework would be inaccurate estimation of the diagnostic matrix $H$ or its spectrum, suggesting that the approximate posterior is more accurate than it truly is. However, computing the eigenvalues or trace of a symmetric matrix, even one estimated from samples, is a well studied problem. And numerical software guards against poor eigenvalue estimation or at least warns if this occurs. We believe the theoretical underpinnings of this work make it robust to undetected failure.

**Does the task/method leverage biases in the data?**    We don't believe our method leverages data bias. As a method for variational inference, our goal is to accurately approximate a posterior distribution. It is very possible to encode biases for/against a particular result in a Bayesian inference problem, but that occurs at the level of modeling (choosing the prior, defining the likelihood) and collecting data, not at the level of approximating the posterior.

## Acknowledgments and Disclosure of Funding

This work was supported in part by the US Department of Energy, Office of Advanced Scientific Computing Research, AEOLUS (Advances in Experimental Design, Optimal Control, and Learning for Uncertain Complex Systems) project. The authors also gratefully acknowledge support from the Inria associate team UNQUESTIONABLE.

## Footnotes

[3]In general $T$ does not need to be a diffeomorphism, but only a particular invertible map; see Appendix B for more details. The distributions we will consider in this paper, however, fulfill the necessary conditions for $T$ to be differentiable almost everywhere.

[4]Code for the numerical examples can be found at `https://github.com/MichaelCBrennan/lazymaps` and `http://bit.ly/2QlelXF`. Data for §4.4, G.4, and G.5 can be downloaded at `http://bit.ly/2XO9Ns8`, `http://bit.ly/2HytQc0` and `http://bit.ly/2Eug5ZR`.

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
