[Supplementary Material]

# A  Proofs

## A.1  Proof of Proposition 1

We first show that for any $T \in \mathcal{T}_r(U)$, there exists a $f : \mathbb{R}^r \to \mathbb{R}_{>0}$ such that (2) holds. Let $T \in \mathcal{T}_r(U)$. Because $T$ is a diffeomorphism we have $T_\sharp \rho(x) = \rho(T^{-1}(x))\det(\nabla T^{-1}(x))$. The inverse of $T$ is given by

$$T^{-1}(x) = \begin{pmatrix} \tau^{-1}(U_r^\top x) \\ U_\perp^\top x \end{pmatrix},$$

and so

$$\det(\nabla T^{-1}(x)) = \det(\nabla \tau^{-1}(U_r^\top x)).$$

Recalling $\rho(x) \propto \exp(-\frac{1}{2}\|x\|_2^2)$, we have that

$$\rho(T^{-1}(x)) \propto \rho(x) \exp\left(-\frac{1}{2}\|\tau^{-1}(U_r^\top x)\|_2^2 + \frac{1}{2}\|U_r^\top x\|_2^2\right),$$

which yields the result of (2) by defining

$$f(U_r^\top x) = \exp\left(-\frac{1}{2}\|\tau^{-1}(U_r^\top x)\|_2^2 + \frac{1}{2}\|U_r^\top x\|_2^2\right)\det(\nabla \tau^{-1}(U_r^\top x)).$$

Now we show that for any function $f : \mathbb{R}^r \to \mathbb{R}_{>0}$ there exists a lazy map $T \in \mathcal{T}_r(U)$ such that (2) holds. Let $f : \mathbb{R}^r \to \mathbb{R}_{>0}$. Denote by $\rho_r$ (resp. $\rho_\perp$) the density of the standard normal distribution on $\mathbb{R}^r$ (resp. $\mathbb{R}^{d-r}$). Let $\tau : \mathbb{R}^r \to \mathbb{R}^r$ be a map that pushes forward $\rho_r$ to $\pi_r$, where $\pi_r$ is the probability density on $\mathbb{R}^r$ defined by $\pi_r(y_r) \propto f(y_r)\rho_r(y_r)$. Such a map always exists because the support of $\pi_r$ (and of $\rho_r$) is $\mathbb{R}^r$ (see [57] for details). Consider the map $Q : \mathbb{R}^d \to \mathbb{R}^d$ defined by

$$Q(z) = \begin{pmatrix} \tau(z_1, \ldots, z_r) \\ z_\perp \end{pmatrix}.$$

Because $\rho = \rho_r \otimes \rho_\perp$, we have $Q_\sharp \rho(y) = \tau_\sharp \rho_r(y_r)\rho(y_\perp) \propto f(y_r)\rho(y)$. Finally, the lazy map

$$T(z) = U_r \tau(z_1, \ldots, z_r) + U_\perp z_\perp = UQ(z)$$

satisfies

$$T_\sharp \rho(z) = U_\sharp(Q_\sharp \rho)(z) \propto f((U^\top z)_r)\rho(U^\top z) \propto f(U_r^\top z)\rho(z).$$

This concludes the proof.

## A.2  Proof of Relation (3)

We can write

$$\mathcal{D}_{\mathrm{KL}}(\pi \| T_\sharp^\star \rho) = \mathbb{E}_\pi[\log(\pi / T_\sharp^\star \rho)] = \mathbb{E}_\pi[\log(\pi / \rho)] - \mathbb{E}_\pi[\log(T_\sharp^\star \rho / \rho)]$$

$$= \mathcal{D}_{\mathrm{KL}}(\pi \| \rho) - \int \log\left(f^\star(U_r^\top x)\right)\pi(x)\mathrm{d}x$$

$$= \mathcal{D}_{\mathrm{KL}}(\pi \| \rho) - \int \log\left(f^\star(x_r)\right)\pi_r(x_r)\mathrm{d}x_r,$$

where $\pi_r(x_r) = (U_r^\top)_\sharp \pi(x_r)$ is the marginal posterior. To complete the result, we must show that $f^\star(x_r) = \pi_r(x_r)/\rho_r(x_r)$. By definition of $f^\star$ we have

$$f^\star(x_r) = \int \frac{\pi(x_r + x_\perp)}{\rho(x_r + x_\perp)}\rho(x_\perp)\mathrm{d}x_\perp = \int \frac{\pi(x_r + x_\perp)}{\rho_r(x_r)\rho_\perp(x_\perp)}\rho(x_\perp)\mathrm{d}x_\perp = \frac{\int \pi(x_r + x_\perp)\mathrm{d}x_\perp}{\rho_r(x_r)} = \frac{\pi_r(x_r)}{\rho_r(x_r)},$$

which concludes the proof.

## A.3 Proof of Proposition 2

Corollary 1 in [59] allows us to write

$$\mathcal{D}_{\mathrm{KL}}(\pi(x)||f^*(U_r^\top x)\rho(x)) \le \frac{1}{2}\mathrm{Tr}\left[(I_d - U_r U_r^\top)H(I_d - U_r U_r^\top)\right].$$

This result follows from a more general *subspace logarithmic Sobolev inequality*, a result that applies to any given projector $P_r \in \mathbb{R}^{d \times d}$ and bounds expectations of the form

$$\mathbb{E}_\pi\left[h^2 \log\left(\frac{h^2}{\mathbb{E}_\pi[h^2|\sigma(P_r)]}\right)\right],$$

where $\sigma(P_r)$ denotes the $\sigma$-algebra generated by $P_r$ and $h$ is a continuously differentiable function. Here we take $P_r = U_r U_r^\top$, the projector onto the subspace spanned by the first $r$ eigenvectors of $H$. The function $h$ is defined in terms of the likelihood model. (See Theorem 1, Corollary 1, and Example 1 in [59], and their proofs, for details.)

Because $U$ is the matrix containing the first eigenvectors of $H$, we have our final result,

$$\mathrm{Tr}\left[(I_d - U_r U_r^\top)H(I_d - U_r U_r^\top)\right] = \lambda_{r+1} + \ldots + \lambda_d.$$

## A.4 Proof of Proposition 3

We define

$$R_\ell = \mathcal{D}_{\mathrm{KL}}((U_r^\ell)^\top)_\sharp \pi_{\ell-1}||\rho_r).$$

Replacing $\pi$ by $\pi_{\ell-1}$ in (3) allows us to write $\mathcal{D}_{\mathrm{KL}}(\pi_{\ell-1}||(T_\ell)_\sharp \rho) = \mathcal{D}_{\mathrm{KL}}(\pi_{\ell-1}||\rho) - R_\ell$ so that

$$\mathcal{D}_{\mathrm{KL}}(\pi_\ell||\rho) = \mathcal{D}_{\mathrm{KL}}(\pi||\rho) - \sum_{k=1}^{\ell-1} R_k.$$

In particular $R_k$ converges to 0 and, because of (5), we have

$$\sup_{\substack{U \in \mathbb{R}^{d \times d} \\ \text{s.t.} UU^\top = I_d}} \mathcal{D}_{\mathrm{KL}}((U_r^\top)_\sharp \pi_{\ell-1}||\rho_r) \xrightarrow[\ell \to \infty]{} 0.$$

By Proposition 14.2 in [28], $\pi_{\ell-1}$ converges weakly to $\rho$. Then $(T_1 \circ \ldots \circ T_\ell)_\sharp \rho$ converges weakly to $\pi$.

# B  Triangular maps

One class of transport maps we consider in our numerical experiments (i.e., to approximate $\tau$ in (1), as a building block within the lazy structure) are lower triangular maps of the form,

$$T(\mathbf{x}) = \begin{bmatrix} T_1(x_1) \\ T_2(x_1, x_2) \\ \vdots \\ T_d(x_1, \ldots, x_d) \end{bmatrix} \tag{7}$$

where each component $T_i$ is monotonically increasing with respect to $x_i$. We will identify these transports with the set $\mathcal{T}_> = \{T : \mathbb{R}^d \to \mathbb{R}^d \,|\, T \text{ is triangular and } \partial_{x_i} T_i > 0\}$. For any two distributions $\rho$ and $\pi$ on $\mathbb{R}^d$ that admit densities with respect to Lebesgue measure (also denoted by $\rho$ and $\pi$, respectively) there exists a unique transport $T \in \mathcal{T}_>$ such that $T_\sharp \rho = \pi$. This transport is known as the Knothe–Rosenblatt (KR) rearrangement [9, 11, 33, 47]. Because $T$ is invertible, the density of the *pullback* measure $T^\sharp \pi$ is given by $T^\sharp \pi(\boldsymbol{x}) = \pi \circ T(\boldsymbol{x}) \det \nabla T(\boldsymbol{x})$, where $\det \nabla T(\boldsymbol{x})$ is defined by $\prod_{i=1}^d \partial_{x_i} T^{(i)}(\boldsymbol{x}_{1:i})$. We note here that $\det \nabla T(\boldsymbol{x})$ is defined formally. Indeed, $T$ does not need to be differentiable (in fact, $T$ inherits the same regularity as the densities of $\rho$ and $\pi$ [9, 50]). In §4.4, and in the additional examples of Appendix F, we consider semi-parametric polynomial approximations to maps in $\mathcal{T}_>$. Specifically, we consider the set $\mathcal{T}_>^\dagger \subset \mathcal{T}_>$ of maps $T : (\mathbf{a}, \boldsymbol{x}) \mapsto T[\mathbf{a}](\boldsymbol{x})$ defined by

$$T_i[\mathbf{c}_i, \mathbf{h}_i](\boldsymbol{x}) := c[\mathbf{c}_i](\boldsymbol{x}_{1:i-1}) + \int_0^{x_i} (h[\mathbf{h}_i](\boldsymbol{x}_{1:i-1}, t))^2 \, dt \,, \tag{8}$$

where $\mathbf{a} = \{(\mathbf{c}_i, \mathbf{h}_i)\}_{i=1}^d$ denotes the coefficients of polynomials $c$ and $h$. As discussed in §2, we compute the transport map (i.e., an approximation to the KR rearrangement) between $\rho$ and $\pi$ as a minimizer $T^\star$ of

$$\min_{T \in \mathcal{T}_>^\dagger} \mathcal{D}_{\mathrm{KL}}(T_\sharp \rho \| \pi) = \min_{T \in \mathcal{T}_>^\dagger} \mathbb{E}_\rho[\log \rho / T^\sharp \pi].$$

[8, 38, 40, 52] provide more details and discussion.

## C   Inverse autoregressive flows

Another underlying class of transports that we use in our numerical experiments are inverse autoregressive flows (IAFs). Introduced in [31], IAFs are a class of normalizing flows parameterized using neural networks. IAFs are built as a composition of component-wise affine transformations, where the shift and scaling functions of each component only depend on earlier indexed variables. Each component of such a transformation can be expressed as

$$T_i(x) = m_i(x_1, \ldots x_{i-1}) + s_i(x_1, \ldots x_{i-1}) x_i$$

where the functions $m_i$ and $s_i$ are defined by neural networks. These maps are naturally lower triangular, and the Jacobian determinant is given by the product of the scaling functions of each component, i.e.,

$$\det(\nabla T) = \prod_{i=1}^d s_i(x),$$

allowing for efficient computation. Flows are typically comprised of several IAF stages with the components either randomly permuted or, as we choose, reversed in between each stage. For the results of §4.2 and §4.3 we construct IAFs using 4 stacked IAF layers. The autoregressive networks each use 2 hidden layers with hidden dimension equal to the *active dimension* of the map (i.e. $d$ in the non-lazy case and $r$ in the lazy case, unless specified) and ELU activation functions. Each map was trained using Adam [30] with step size $10^{-3}$ for 20000 iterations. The optimization objective (i.e., the ELBO) was approximated using 100 independent samples from $\rho$ at each iteration.

## D   Generalized linear models and lazy structure

Here we discuss how generalized linear models may naturally admit lazy structure. We consider a Bayesian logistic regression problem as an example, but the same result follows for other generalized linear models. Let $M$ denote the number of observations in a data set and $N$ denote the number of covariates or features. In §4.2, we considered $N = 500$ covariates. The low rank problem used $M = 20$ observations and the full rank problem used $M = 605$ observations. For each observation $i = 1, \ldots, M$ and covariate $j = 1, \ldots, N$, we denote the observed covariates by $f_{ij} \in \mathbb{R}$, the observations as $y_i \in \{0, 1\}$, and the model parameters as $x_j \in \mathbb{R}$. The single observation likelihood is then defined as

$$\ell_i(\mathbf{x}) = P(\mathbf{x}, \mathbf{f}_i)^{y_i} (1 - P(\mathbf{x}, \mathbf{f}_i))^{1-y_i}$$

where the quantity

$$P(\mathbf{x}, \mathbf{f}_i) = \left(1 + \exp(-\mathbf{x}^T \mathbf{f}_i)\right)^{-1} = \mathrm{sigmoid}(\mathbf{x}^T \mathbf{f}_i)$$

models the probability that $y_i = 1$. This has the form of a generalized linear model, i.e., the likelihood depends on a linear function of the covariates, $\mathbf{x}^T \mathbf{f}_i$. The gradient of the log likelihood then has the form

$$\nabla_{\mathbf{x}} \log(\ell_i(\mathbf{x})) = \mathbf{f}_i h(\mathbf{x}; \mathbf{f}_i, y_i)$$

for some function $h$. Assuming independence of the observations, the likelihood of the data set can be written as

$$\nabla_{\mathbf{x}} \log(\mathcal{L}(\mathbf{x})) = \sum_{i=1}^M \mathbf{f}_i h(\mathbf{x}; \mathbf{f}_i, y_i) = \mathbf{F} \mathbf{h}(\mathbf{x}).$$

The matrix $\mathbf{F}$ is often referred to as the design matrix. We can then express the diagnostic matrix $H$ as

$$H = \int \left(\nabla \log(\mathcal{L}(\mathbf{x}))\right) \left(\nabla \log(\mathcal{L}(\mathbf{x}))\right)^T d\pi = \mathbf{F} \left[\int (\mathbf{h}(\mathbf{x})) (\mathbf{h}(\mathbf{x}))^T d\pi\right] \mathbf{F}^T,$$

and so the rank of $H$ is bounded by the rank of the feature matrix $\mathbf{F}$ which is at most $\min(N, M)$. If $M < N$, we are in the exactly lazy setting, where $r = M$. We also note that $\mathbf{F}$ may be low rank due to redundancy in the measurements, meaning when $\mathbf{f}_i$ is nearly aligned with $\mathbf{f}_j$; more generally, it might exhibit some spectral decay.

# E    The use of $H^{\mathbf{B}}$ vs $H$

We note in §2 that a practical implementation of Algorithm 1 requires the numerical approximation of the diagnostic matrix $H$ defined by

$$H = \int \left(\nabla \log \frac{\pi}{\rho}\right) \left(\nabla \log \frac{\pi}{\rho}\right)^{\top} d\pi.$$

This poses a challenge as we cannot generate samples from $\pi$. We can obtain an (asymptotically) unbiased estimate of $H$ using self-normalized important sampling (IS), but as we comment in the main text, this estimate typically has large variance when the IS instrumental/biasing distribution is far from $\pi$. Instead, we can use the diagnostic matrix $H^{\mathbf{B}}$, where the expectation is instead taken with respect to the reference density $\rho$

Figure 4: The two trace diagnostics through out the training of $U_r$-IAF on the low rank Bayesian logistic regression problem .

$$H^{\mathbf{B}} = \int \left(\nabla \log \frac{\pi}{\rho}\right) \left(\nabla \log \frac{\pi}{\rho}\right)^{\top} d\rho.$$

Unbiased estimates of $H^{\mathbf{B}}$ can be computed easily using direct Monte Carlo sampling, but these are of course biased estimates of $H$ in general. In this section we comment on the use of this biased estimate in the error bound on the KL divergence, and find that this bias leads to a more conservative diagnostic.

Figure 5a shows histograms of 100 estimates of $\mathrm{Tr}(\widehat{H})$ (where $\widehat{H}$ is a self-normalized IS estimate of $H$) and $\mathrm{Tr}(\widehat{H}^{\mathbf{B}})$ (where $\widehat{H}^{\mathbf{B}}$ is a Monte Carlo estimate of $H^{\mathbf{B}}$) for the low-rank logistic regression problem of §4.2. Each estimate was constructed from $K = 500$ samples. We see that the variance of $\mathrm{Tr}(\widehat{H})$ is higher than that of $\mathrm{Tr}(\widehat{H}^{\mathbf{B}})$. Figure 5b shows similar histograms for $\mathrm{Tr}(\widehat{H}_\ell)$ and $\mathrm{Tr}(\widehat{H}_\ell^{\mathbf{B}})$ after the training of the transport map. We see that the bias has decreased now that the approximate posterior is close to the true posterior; where indeed $H_\ell^{\mathbf{B}}$ is closer to $H_\ell$. The variance of the IS estimate $\mathrm{Tr}(\widehat{H}_\ell)$ has decreased significantly as well. Figure 4 shows the two trace diagnostics computed throughout the training of the $U_r$-IAF lazy map. We see that $\frac{1}{2}\mathrm{Tr}(H_\ell^{\mathbf{B}}) > \frac{1}{2}\mathrm{Tr}(H_\ell)$ throughout the training process, meaning it is a more conservative error bound for this particular problem.

# F    Numerical algorithms

Here we describe the numerical algorithms required by the lazy map framework. Algorithm 3 assembles the numerical estimate $\widehat{H}^{\mathbf{B}}$ via some quadrature rule (e.g. Monte Carlo, Gauss quadrature [25], sparse grids [51], ect.) of $H^{\mathbf{B}} = \int (\nabla \log \frac{\pi}{\rho})(\nabla \log \frac{\pi}{\rho})^{\top} d\rho$.

Algorithm 4 computes the eigenvectors $U$ satisfying Proposition 2 and discerns between the subspace of relevant directions $\mathrm{span}(U_r)$ and its orthogonal complement $\mathrm{span}(U_\perp)$.

Algorithm 5 outlines the numerical solution of the variational problem

$$T[\mathbf{a}^{\star}] = \arg \min_{T[\mathbf{a}] \in \mathcal{T}} \mathcal{D}_{\mathrm{KL}} \left(T[\mathbf{a}]_{\sharp}\rho \| \pi\right). \tag{9}$$

For the sake of simplicity we fix the complexity the underlying transport class $\mathcal{T}$ and the sample size $m$ used in the discretization of the KL divergence. Alternatively one could adaptively increase the complexity and the sample size to match a prescribed tolerance, following the procedure described in [8]. For the examples presented in this work, the variational problem is solved either with the Adam

(a) Before training            (b) After training

Figure 5: Histograms of $\mathrm{Tr}(H^{\mathrm{B}})$ and $\mathrm{Tr}(H)$ before and after training for the low rank logistic regression problem.

optimizer [30] or with the Broyden–Fletcher–Goldfarb–Shanno (BFGS) quasi-Newton method [10]. One could switch to a full Newton method if the Hessian of $\pi$ or its action on a vector are available.

Algorithms 6 and 7 are numerical counterparts of Algorithms 1 (constructing a lazy map) and 2 (constructing a deeply lazy map) respectively.

---

**Algorithm 3** Given the quadrature rule $(x_i, w_i)_{i=1}^m$ with respect to the base distribution $\rho$, and the unnormalized density $\pi$, compute an approximation to $H^{\mathrm{B}} = \int (\nabla \log \frac{\pi}{\rho})(\nabla \log \frac{\pi}{\rho})^\top \mathrm{d}\rho$.

---

1: **procedure** COMPUTEH( $(x_i, w_i)_{i=1}^m, \pi$ )
2:      Assemble

$$\widehat{H}^{\mathrm{B}} = \sum_{i=1}^m \left( \nabla_{\mathbf{x}} \log \frac{\pi(x_i)}{\rho(x_i)} \right) \left( \nabla_{\mathbf{x}} \log \frac{\pi(x_i)}{\rho(x_i)} \right)^T w_i$$

     **return** $\widehat{H}^{\mathrm{B}}$
3: **end procedure**

---

---

**Algorithm 4** Given the matrix $\widehat{H}^{\mathrm{B}} \approx H^{\mathrm{B}}$, the tolerance $\varepsilon$, and a maximum lazy rank $r_{\max}$, find the matrix $U \coloneqq [U_r \,|\, U_\perp]$ that satisfies Proposition 2.

---

1: **procedure** COMPUTESUBSPACE( $\widehat{H}^{\mathrm{B}}, \varepsilon, r_{\max}$ )
2:      Solve the eigenvalue problem $\widehat{H}^{\mathrm{B}} X = \Lambda X$
3:      Let $r = r_{\max} \wedge \min\{r \le d \,: \frac{1}{2} \sum_{i>r} \lambda_i \le \varepsilon\}$
4:      Define $U_r = [X_{:,1}, \ldots, X_{:,r}]$ and $U_\perp = [X_{:,r+1}, \ldots, X_{:,n}]$
5:      **return** $U_r, U_\perp, r$
6: **end procedure**

---

**Algorithm 5** Given the quadrature rule $(x_i, w_i)_{i=1}^m$ with respect to the base distribution $\rho$, the unnormalized target density $\pi$, a set of underlying class of transport maps $\mathcal{T}$, a tolerance $\varepsilon_{\text{map}}$, find the optimal map parameters $\mathbf{a}^\star$ such that $T[\mathbf{a}]_\sharp \rho \propto \pi$ by minimizing (9).

1: **procedure** COMPUTEMAP( $(x_i, w_i)_{i=1}^m$, $\pi$, $\mathcal{T}_{\mathbf{a}}$, $\varepsilon_{\text{map}}$ )
2:     Solve (e.g., via a stochastic or deterministic optimization method),

$$T[\mathbf{a}^\star] = \underset{T[\mathbf{a}] \in \mathcal{T}}{\arg\min} \underbrace{- \sum_{i=1}^m \log(T[\mathbf{a}]^\sharp \pi(x_i)) w_i}_{\mathcal{J}[\mathbf{a}]} ,$$

based on some stopping criteria, e.g., $\quad \|\nabla_{\mathbf{a}} \mathcal{J}[\mathbf{a}^\star]\|_2 < \varepsilon_{\text{map}}$

3:     **return** $T[\mathbf{a}^\star]$
4: **end procedure**

---

**Algorithm 6** Given the quadrature rule $(x_i, w_i)_{i=1}^m$ with respect to the base distribution $\rho$, the unnormalized density $\pi$, the matrix $\widehat{H}^{\text{B}} \approx H^{\text{B}}$, the rank truncation tolerance $\varepsilon_{\text{r}}$, the maximum lazy rank $r_{\max}$, the class of transport maps $\mathcal{T}$ and the target tolerance $\varepsilon_{\text{map}}$ for learning the map $\tau$, identify the optimal lazy map $T$.

1: **procedure** LAZYMAPCONSTRUCTION( $(x_i, w_i)_{i=1}^m$, $\pi$, $\widehat{H}^{\text{B}}$, $\varepsilon_{\text{r}}$, $r_{\max}$, $\mathcal{T}$, $\varepsilon_{\text{map}}$, )
2:     $U_r, U_\perp, r \leftarrow$ COMPUTESUBSPACE( $\widehat{H}^{\text{B}}$, $\varepsilon_{\text{r}}$, $r_{\max}$ )          $\triangleright$ Algorithm 4
3:     Define $\hat{\pi}(x) := (U_r|U_\perp)^\sharp \pi(x) = \pi \circ (U_r|U_\perp) x$
4:     Build the quadrature $(x_i, w_i)_{i=1}^m$ with respect to $\mathcal{N}(0, I_d)$
5:     Define $\mathcal{T}_r = \left\{ T[\mathbf{a}](z) = [\tau[\mathbf{a}](z_1, \ldots, z_r)^\top, z_{r+1}, \cdots, z_d]^\top \; \middle| \; \tau[\mathbf{a}] \in \mathcal{T} \right\}$
6:     $T[\mathbf{a}^\star] \leftarrow$ COMPUTEMAP( $(x_i, w_i)_{i=1}^m$, $\hat{\pi}$, $\mathcal{T}_r$, $\varepsilon_{\text{map}}$ )          $\triangleright$ Algorithm 5
7:     Define $L(z) := (U_r|U_\perp) \circ T[\mathbf{a}](z)$
8:     **return** $L$
9: **end procedure**

---

**Algorithm 7** Given the quadrature rule $(x_i, w_i)_{i=1}^m$ with respect to the base distribution $\rho$, the target density $\pi$, a stopping tolerance $\varepsilon$ and a maximum number of lazy layers $\ell_{\max}$, compute a deeply lazy map. See Algorithm 6 for the definition of the remaining arguments.

1: **procedure** LAYERSOFLAZYMAPSCONSTRUCTION($(x_i, w_i)_{i=1}^m$, $\pi$, $\varepsilon$, $r$, $\ell_{\max}$, $\mathcal{T}$, $\varepsilon_{\text{map}}$)
2:     Set $\pi_0 = \pi$ and $\ell = 0$
3:     Build the quadrature $(x_i, w_i)_{i=1}^m$ with respect to $\mathcal{N}(0, I_d)$
4:     Compute $\widehat{H}_\ell^{\text{B}} =$ COMPUTEH( $(x_i, w_i)_{i=1}^m$, $\pi_\ell$ )
5:     **while** $\ell \leq \ell_{\max}$ and $\frac{1}{2} \operatorname{Tr}(\widehat{H}_\ell^{\text{B}}) \geq \varepsilon$ **do**
6:         $\ell \leftarrow \ell + 1$
7:         $T_\ell \leftarrow$ LAZYMAPCONSTRUCTION( $(x_i, w_i)_{i=1}^m$, $\pi_{\ell-1}$, $\widehat{H}_\ell^{\text{B}}$, $0$, $r$, $\mathcal{T}$, $\varepsilon_{\text{map}}$ ) $\triangleright$ Algorithm 6
8:         Update $\mathfrak{T}_\ell = \mathfrak{T}_{\ell-1} \circ T_\ell$
9:         Compute $\pi_\ell = (\mathfrak{T}_\ell)^\sharp \pi$
10:         Build the quadrature $(x_i, w_i)_{i=1}^m$ with respect to $\mathcal{N}(0, I_d)$
11:         Compute $\widehat{H}_\ell^{\text{B}} =$ COMPUTEH( $(x_i, w_i)_{i=1}^m$, $\pi_\ell$ )
12:     **end while**
13:     **return** $\mathfrak{T}_\ell = T_1 \circ \cdots \circ T_\ell$
14: **end procedure**

# G   Numerical examples: additional details and experiments

In this section, we provide more details concerning our numerical examples and present several other numerical experiments.

## G.1   Additional details: Bayesian logistic regression

Here we provide addition details and results for the Bayesian logistic regression problems discussed in §4.2. We begin by further describing the UCI Parkinson's disease data set [1]. The $500$ features we consider consist of the patient sex, and audio extensions from a patient recording. The data set includes data from 3 independent recordings from $188$ Parkinson's disease patients and a control group of $64$ individuals, totaling $756$ observations in all. The low rank problem considers $20$ observations where we use observations from 20 different individuals.

We imposed a non-informative prior of $\mathcal{N}(0, 10^2 I_d)$ on the parameters. Samples from the prior can be transformed to match those of a standard normal distribution via a *whitening* transformation, i.e.

$$z \sim \mathcal{N}(0, 10^2 I_d), \ W z := \frac{1}{10} z \sim \mathcal{N}(0, I_d),$$

where we let $W$ denote this whitening operation. We consider the transformed posterior

$$\widetilde{\pi}(x) \propto \mathcal{L}(W^{-1}x)\rho(x)$$

where the prior has been replaced with a standard normal distribution. This whitened posterior relates to the true posterior by $\widetilde{\pi} = W_\sharp \pi$. We see that solving this transformed problem is equivalent to solving the original, and that working with this whitened problem directly exposes lazy structure by matching the form of 2. A similar whitening process is followed for each of the numerical experiments.

Figure 6 shows mean performance metrics through out the training process for each of the maps considered. Each metric is computed with $500$ independent samples. For G3-IAF, the three lazy layers were trained for 5000, 5000 and 10000 iterations, which can be seen as sharp decreases in the negative ELBO and trace diagnostics occur. In general we see faster convergence in terms of the number of iterations for maps using the lazy framework compared to the baselines.

## G.2   Additional details: Bayesian neural network

In §4.3 we considered a Bayesian neural network that is also used as a test problem in [36] and [18]. Bayesian neural networks generate high dimension inference problems, where the parameter dimension is the number of parameters in the underlying neural network. We considered the UCI yacht hydrodynamics data set [2]. In our example, the parameter dimension is $581$, given an input dimension of 6, one hidden layer of dimension 20, and output layer of dimension 1. We use sigmoid activation functions in the input and hidden layer, and a linear output layer. The prior on the model parameters is taken to be zero mean Gaussian with a variance of $100$.

Here we consider affine maps, i.e., maps of the form $T(x) = \mu + Lx$, where $L \in \mathbb{R}^{d \times d}$ denotes a lower triangular matrix and $\mu \in \mathbb{R}^d$ a constant vector. The approximate posteriors in this case are indeed Gaussian distributions with mean $\mu$ and covariance $\Sigma = LL^T$. We note that the final approximate posterior given by the G3-affine transport map is also Gaussian given that the composition of affine functions is affine. Therefore the performance benefits we see may come from avoiding sub-optimal minima of the KL divergence. We see stabler training in terms of the performance metrics in Figure 7. For G3-affine, layers were trained for 5000, 5000 and 10000 iterations, where we see sharp decreases in each of the diagnostics.

## G.3   Additional details: High-dimensional elliptic PDE inverse problem

Here we explain how the numerical discretization of the PDE enters the Bayesian inference formulation. We denote by $\mathcal{S}$ the map $\boldsymbol{\kappa} \mapsto u$, mapping the discretized coefficient to the numerical solution of equation 6. The observation map is defined by the operator $B_i(u) := \int_{\mathcal{D}} u\,\phi_i\,\mathrm{d}\boldsymbol{x}$, where $\phi_i(\boldsymbol{x}) := \exp[-\|\boldsymbol{s}_i - \boldsymbol{x}\|_2^2/(2\delta^2)]/\gamma_i$, $\{\boldsymbol{s}_i\}_{i=1}^n \in \mathcal{D}$ are observation locations, $\delta = 10^{-4}$, and $\gamma_i$ are normalization constants so that $\int_{\mathcal{D}} \phi_i\,\mathrm{d}\boldsymbol{x} = 1$ for all $i = 1, \ldots, n$. The parameter-to-observation map

(a) Low rank - Negative ELBO

(b) Full rank - Negative ELBO

(c) Low rank - $\mathrm{Tr}(H^{\mathrm{B}})$

(d) Full rank - $\mathrm{Tr}(H^{\mathrm{B}})$

(e) Low rank - $\mathrm{Tr}(H)$

(f) Full rank - $\mathrm{Tr}(H)$

Figure 6: Mean training plots for the low rank (left) and full rank (right) Bayesian logistic regression problems across 10 optimization runs. *The $x$-axes include the cost of forming the matrices $H_\ell$ to determine the subspace $U^\ell$ in terms of gradient evaluations. Each matrix is computed using 500 gradient evaluations, the same cost as 5 optimization steps.

(a) Negative ELBO

(b) $\mathrm{Tr}(H^{\mathrm{B}})$

(c) $\mathrm{Tr}(H)$

(d) Spectrum of $H_\ell^{\mathrm{B}}$ before and after training each lazy layer.

Figure 7: (a,b,c) Mean training plots for the Bayesian neural network problem across 10 optimization runs. *The $x$-axes include the cost of forming the matrices $H_\ell$ to determine the subspace $U^\ell$ in terms of gradient evaluations. Each matrix is computed using 581 gradient evaluations, approximately the same cost as 6 optimization steps. (d) Analogous plot to Figure 2 for the Bayesian neural network problem. The spectrum of the diagnostic matrices $H_\ell^{\mathrm{B}}$ flatten and fall as the approximation to the posterior improves with each lazy layer.

(a) Solution $\mathcal{S}(\boldsymbol{\kappa}^\star)$

(b) $\mathbb{E}[\boldsymbol{\kappa}|\boldsymbol{y}^\star]$

(c) $\mathrm{Std}[\boldsymbol{\kappa}|\boldsymbol{y}^\star]$

Figure 8: Additional figures for the elliptic problem with unknown diffusion coefficient. Figure (a) shows the solution $u$ corresponding to the field in Figure 3a. Figures (b) and (c) show the mean and the standard deviation of the posterior distribution.

is then defined by $\mathcal{F} : \boldsymbol{\kappa} \mapsto [B_1(\mathcal{S}(\kappa)), \ldots, B_n(\mathcal{S}(\kappa))]^\top$. The coefficient $\kappa$ is endowed with the distribution $\kappa \sim \mathcal{N}(0, \mathcal{C}(\boldsymbol{x}, \boldsymbol{x}'))$, where $\mathcal{C}(\boldsymbol{x}, \boldsymbol{x}') := \exp(-\|\boldsymbol{x} - \boldsymbol{x}'\|_2)$ is the Ornstein–Uhlenbeck (exponential) covariance kernel. Letting $\Sigma$ be the discretization of $\mathcal{C}$ over the finite element mesh, we define the likelihood to be $\mathcal{L}_{\boldsymbol{y}}(\boldsymbol{z}) \propto \exp\left( -\|\boldsymbol{y} - \mathcal{F}(\Sigma^{1/2}\boldsymbol{z})\|_{\Sigma_{\text{obs}}^{-1}} \right)$. We stress here that the model is computationally demanding: the evaluation of $\pi(\boldsymbol{z})$ and $\nabla\pi(\boldsymbol{z})$ require approximately 1 second.

Figure 8 shows the observation generating solution $u^\star = \mathcal{S}(\boldsymbol{\kappa})$, the posterior mean $\mathbb{E}[\boldsymbol{\kappa}|\boldsymbol{y}^\star]$ and the posterior standard deviation $\text{Std}[\boldsymbol{\kappa}|\boldsymbol{y}^\star]$.

### G.4 Additional example: Log-Gaussian Cox process with sparse observations

We consider an inference problem in spatial statistics for a log-Gaussian Cox point process on a square domain $\mathcal{D} = [0, 1]^2$. This type of stochastic process is frequently used to model spatially aggregated point patterns [14, 24, 39, 48]. Following a configuration similar to [14, 39], we discretize $\mathcal{D}$ into a $64 \times 64$ uniform grid, and denote by $\boldsymbol{s}_i \in \mathcal{D}$ the center of the $i$th cell, for $i = 1, \ldots, d$, with $d = 64^2$. We consider a discrete stochastic process $(\boldsymbol{Y}_i)_{i=1}^d$, where $\boldsymbol{Y}_i$ denotes the number of occurrences/points in the $i$th cell. Each $\boldsymbol{Y}_i$ is modeled as a Poisson random variable with mean $\exp(\boldsymbol{Z}_i)/d$, where $(\boldsymbol{Z}_i)$ is a Gaussian process with covariance $\text{Cov}(\boldsymbol{Z}_i, \boldsymbol{Z}_j) = \sigma^2 \exp\left( -\|\boldsymbol{s}_i - \boldsymbol{s}_j\|_2 / (64\beta) \right)$ and mean $\mathbb{E}[\boldsymbol{Z}_i] = \mu$, for all $i = 1, \ldots, d$. We consider the following values for the parameters: $\beta = 1/33$, $\sigma^2 = 1.91$, and $\mu = \log(126) - \sigma^2/2$. The $(\boldsymbol{Y}_i)$ are assumed conditionally independent given the (latent) Gaussian field. For interpretability reasons, we also define the *intensity* process $(\boldsymbol{\Lambda}_i)_{i=1}^d$ as $\boldsymbol{\Lambda}_i = \exp(\boldsymbol{Z}_i)$, for $i = 1, \ldots, d$.

The goal of this problem is to infer the posterior distribution of the latent process $\boldsymbol{\Lambda} := (\boldsymbol{\Lambda}_1, \ldots, \boldsymbol{\Lambda}_n)$ given few sparse realizations of $\boldsymbol{Y} := (\boldsymbol{Y}_i)$ at $n = 30$ spatial locations $\boldsymbol{s}_{k_1}, \ldots, \boldsymbol{s}_{k_n}$ shown in Figure 9a. We denote by $\boldsymbol{y}^\star \in \mathbb{R}^n$ a realization of $\boldsymbol{Y}$ obtained by sampling the latent Gaussian field according to its marginal distribution. Our target distribution is then: $\pi_{\boldsymbol{\Lambda}|\boldsymbol{Y}}(\boldsymbol{\lambda}|\boldsymbol{y}^\star)$.

Since the posterior is nearly Gaussian, we will run three experiements with linear lazy maps and ranks $r = 1, 3, 5$. For the three experiments, the KL-divergence minimized for each lazy layer and the estimators of $H_\ell^{\text{B}}$ are discretized with $m = 100, 300, 500$ Monte Carlo samples respectively.

Figures 9b–c show the expectation and few realizations of the posterior, confirming the data provides some valuable information to recover the field $\boldsymbol{\Lambda}$. Figures 9d–e show the convergence rate and the cost of the algorithm as new layers of lazy maps are added to $\mathfrak{T}_\ell$. As we expect, the use of maps with higher ranks leads to faster convergence. On the other hand the computational cost per step increases—also due to the fact that we increase the sample size $m$ as the rank increases. Figure 9f reveals the spirit of the algorithm: each lazy map trims away power from the top of the spectrum of $H$, which slowly flattens *and* decreases. To additionally confirm the quality of $\mathfrak{T}_6$ for lazy maps with rank 5, and to produce asymptotically unbiased samples from $\pi$, we sample the pullback distribution $\mathfrak{T}_6^\sharp \pi$ using an MCMC chain of length $10^4$, with a Metropolis independence sampler employing a $\mathcal{N}(0, I_d)$ proposal (see [C. Robert and G. Casella, *Monte Carlo statistical methods*, 2013] for more details). As explained in [44], the Metropolis independence sampler is effective insofar as the pullback distribution has been Gaussianized by the map. The reported acceptance rate is $72.6\%$ with the worst effective sample size (over all $d = 4096$ chain components) being $26.6\%$ of the total chain.

### G.5 Additional example: Estimation of the Young's modulus of a cantilever beam

Here we consider the problem of estimating the Young's modulus $E(x)$ of an inhomogeneous cantilever beam, i.e., a beam clamped on one side ($x = 0$) and free on the other ($x = l$). The beam has a length of $l = 10\,\text{m}$, a width of $w = 10\,\text{cm}$ and a thickness of $h = 30\,\text{cm}$. Using Timoshenko's beam theory, the displacement $u(x)$ of the beam under the load $q(x)$ is modeled by the coupled PDEs

$$\begin{cases} \frac{\mathrm{d}}{\mathrm{d}x}\left[ \frac{E(x)}{2(1+\nu)} \left( \varphi(x) - \frac{\mathrm{d}}{\mathrm{d}x}w(x) \right) \right] = \frac{q(x)}{\kappa A} \,, \\ \frac{\mathrm{d}}{\mathrm{d}x}\left( E(x)I\frac{\mathrm{d}}{\mathrm{d}x}\varphi(x) \right) = \kappa A \frac{E(x)}{2(1+\nu)} \left( \varphi(x) - \frac{\mathrm{d}}{\mathrm{d}x}w(x) \right) \,, \end{cases} \tag{10}$$

where $\nu = 0.28$ is the Poisson ratio, $\kappa = 5/6$ is the Timoshenko shear coefficient for rectangular sections, $A = wh$ is the cross-sectional area of the beam, and $I = wh^3/12$ is its second moment of inertia. We consider a beam composed of $d = 5$ segments each of $2\,\text{m}$ length made of different kinds of steel, with Young's moduli $\boldsymbol{E}^\star = \{E_i\}_{i=1}^5 = \{190, 213, 195, 208, 200\,\text{GPa}\}$ respectively, and we

(a) Field $\boldsymbol{\Lambda}^\star$ and observations $\boldsymbol{y}^\star$     (b) $\mathbb{E}[\boldsymbol{\Lambda}|\boldsymbol{y}^\star]$     (c) Realizations of $\boldsymbol{\Lambda} \sim \pi_{\boldsymbol{\Lambda}|\boldsymbol{y}^\star}(\boldsymbol{\lambda})$

(d) Convergence rate     (e) Convergence and cost vs. ranks     (f) Spectrum decay

Figure 9: Application of the algorithm on the log-Gaussian Cox process distribution. Figure (a) shows the intensity field $\boldsymbol{\Lambda}^\star$ used to generate the data $\boldsymbol{y}^\star$ (circles). Figures (b) shows the posterior expectation. Figure (c) shows four realizations from the posterior $\pi(\boldsymbol{\Lambda}|\boldsymbol{y}^\star)$. Figure (d) shows the convergence rate of the algorithm as a function of the iterations. Figure (e) shows the cost of the algorithm for different truncation ranks. Figure (f) shows the decay of the spectrum of $H_\ell$ for lazy maps with rank 5.

(a) Experimental setting     (b) Convergence rate     (c) Marginals of $\pi(\boldsymbol{E}|\boldsymbol{y}^\star)$

Figure 10: Application of the algorithm for the estimation of the Young's modulus of a cantilever beam. Figure (a) shows the experimental setting with the beam clamped at $x = 0$, the load applied at $x = l$, 20 sensors marked in red, and the true Young's modulus [GPa] for each segment. Figure (b) shows the convergence of the algorithm in terms of the variance and trace diagnostics. Figure (c) shows marginals of the posterior distribution $\pi(\boldsymbol{E}|\boldsymbol{y}^\star)$ along with the true values (red).

run the virtual experiment of applying a point mass of $5\,\mathrm{kg}$ at the tip of the beam. Observations $\boldsymbol{y}^\star$ of the displacement $w$ are gathered at the locations shown in Figure 10a with a measurement noise of $1\,\mathrm{mm}$. We endow $\boldsymbol{E}$ with the prior $\pi(\boldsymbol{E}) = \mathcal{N}(200\,\mathrm{GPa}, 25 \cdot I_5)$ and our goal is to characterize the posterior distribution $\pi(\boldsymbol{E}|\boldsymbol{y}^\star) \propto \mathcal{L}_{\boldsymbol{y}^\star}(\boldsymbol{E})\pi(\boldsymbol{E})$. Let $\mathcal{S}$ be the map $\boldsymbol{E} \mapsto w$ delivering the solution to (10). Observations are gathered through the operator $B_i(w) := \int_0^l w\,\phi_i\,\mathrm{d}\boldsymbol{x}$, where $\phi_i$ are defined the same way as in Appendix G.3 for locations $\{s_i := i \cdot 0.5\}_{i=1}^{20}$. Defining the parameter-to-observable map $\mathcal{F} : \boldsymbol{E} \mapsto [B_1(\mathcal{S}(\boldsymbol{E})), \ldots, B_{20}(\mathcal{S}(\boldsymbol{E}))]^\top$, observations $\boldsymbol{y}$ are assumed to satisfy the model $\boldsymbol{y} = \mathcal{F}(\boldsymbol{E}) + \epsilon$, where $\epsilon \sim \mathcal{N}(0, 10^{-6} \cdot I_{20})$ corresponds to $1\,\mathrm{mm}$ of measurement noise.

The algorithm is run with rank 2 lazy maps using triangular polynomial maps of degree 3 as the underlying transport class. The expectations appearing in the algorithms are approximated using

(a) True field vs. posterior      (b) Posterior predictive mismatch

Figure 11: Additional results for the estimation of Young's modulus of a cantilever beam. Figure (a) shows the mean (dashed black) and the $5, 10, 90, 95$-percentiles (thin black) of the marginals of $\pi(\boldsymbol{E}|\boldsymbol{y}^\star)$ compared with the true values (red). Figure (b) shows the distribution of $(\boldsymbol{y}^\star - \boldsymbol{y})/|\boldsymbol{y}^\star|$, where $\boldsymbol{y}$ is distributed according to the posterior predictive $\pi(\boldsymbol{y}|\boldsymbol{y}^\star) = \pi(\boldsymbol{y}|\boldsymbol{E})\pi(\boldsymbol{E}|\boldsymbol{y}^\star)$.

$m = 100$ samples from $\mathcal{N}(0, I_5)$. Figures 10 and 11 summarize the results. We further confirm these results by generating an MCMC chain of length $10^4$ using Metropolis-Hastings with a $\mathcal{N}(0, I_d)$ independence proposal; the target distribution for MCMC is the pullback $\mathfrak{T}_\ell^\sharp \pi$, as in previous examples. The reported acceptance rate is $68.3\%$ with the worst, best, and average effective sample sizes being $7.0\%$, $38.7\%$, and $20.1\%$ of the complete chain. In this example we fix the Poisson ratio, but one could think of it varying from material to material, and thus estimate it jointly with the Young's modulus.