[Reviews · NeurIPS 2020]

Review 1

Summary and Contributions: This paper presents a novel transport-based method for sampling from intractable posterior distributions in the context of Bayesian inference. The method, based on compositions of "lazy" transport maps, first identifies low-dimensional subspaces in the structure of the problem, and then learns a flow-based transport method on these subspaces. It is shown how to derive a computable bound on the KL divergence between the posterior and the pushforward of a simpler distribution by a lazy map. A novel training procedure is also derived, and this procedure is shown to weakly converge to the target in certain settings. The method is shown to outperform baseline transport methods in numerical experiments.

Strengths: The whole framework of lazy maps, including their construction, training procedure, error bounds, and theoretical benefits are all very novel and interesting. That is to say this work feels very original, which is quite refreshing. I find this work to be much more theoretically grounded than a typical normalizing flow paper, inspiring more faith in its ability to generalize to other problems besides those considered in the paper. The experiments have demonstrated to me that (deeply) lazy maps can outperform baseline flow methods in Bayesian inference, and may also provide some benefits for MCMC-based approaches to inference. This paper is very much relevant to the NeurIPS community, as it provides a *variational* method for *Bayesian inference* relying on *normalizing flows*.

Weaknesses: I have some concerns about the correctness or implications Proposition 3, which is one of the main parts of the motivation. I have expanded on this further in the "additional feedback". This method is only described as a method for sampling from intractable targets. Thus it would be worthwhile to compare to MCMC more, beyond just mentioning it in the second-last paragraph of the paper. It is stated that the lazy maps use fewer parameters and converge faster than the baselines. However, we know that deep lazy maps require several optimization rounds to converge. How do the methods compare on wall-clock time? There is no dedicated related work section, which can be very useful even for original contributions.

Correctness: Besides my concerns about Proposition 3, I believe the rest of the paper to be correct in its claims and methodology.

Clarity: The paper is fairly well written and has minimal typos, although there are two points I would like to mention: 1. Again, Proposition 3 is somewhat explained but leaves me with more questions than answers. 2. In section 2, the derivation of a couple very important results (eqs. 3 and 4) is simply deferred to previous work. It would be nice to have these derived at least in the appendix in your own notation/framework, which I'm assuming you had to do anyway to ensure the results translated to your setting. Obviously a paper cannot be entirely self-contained, but including some important derivations which are not totally standard can be useful for readability.

Relation to Prior Work: The introduction touches on some works, and there are some comparisons scattered throughout the paper, but the paper's discussion of prior work overall somewhat unsatisfactory.

Reproducibility: Yes

Additional Feedback: ### POST AUTHOR FEEDBACK ### I am raising my score as the authors have done a good job of addressing my feedback and the other reviews were favourable. ### PRE AUTHOR FEEDBACK ### First of all, I would like to thank the authors for their submission, which I found to be quite interesting and original, as already mentioned. I like the idea of intelligently reducing a higher-dimensional problem to a series of lower-dimensional problems, the adaptive error bounds on the approximation, and the map-learning procedure which involves more than just defining a loss function and blindly optimizing. However, I also have some comments / questions which, if addressed, would very much solidify this paper's contribution to the field of machine learning in my opinion. As mentioned a few times already, I would like some clarity on Proposition 3. Specifically: (### POST FEEDBACK NOTE - I misunderstood on first read, thank you for clarifying in your response.) 1. Isn't the condition (6) essentially vacuous as long as $(U_r^\ell)^T_\# \pi_{\ell-1}$ does not *exactly* equal $\rho_r$, since the KL divergence is positive? I guess this could be considered a good thing for weak convergence, but then why even include this condition? 2. Let's consider the case where $\ell = 1$, $r = 1$, and the dimensionality of the problem is much bigger than 1. I do not understand how $(T_1)#\rho$ has any chance of converging weakly to $\pi$, considering it has only updated one dimension of the map nonlinearly. What is going on in this scenario? Does this construction violate condition (6) somehow? I would also have liked to see a bit more of a comparison made to MCMC methods, since, at its core, this paper focuses on building a method for Bayesian inference, and MCMC is one of the leading approaches for doing this. The paper focuses on how their work improves the *transport* approach to Bayesian inference, but a comparison with MCMC is not made until the second last paragraph. A bit more discussion or experimentation here would be appreciated. This could also be included in a related work section. *** SMALLER COMMENTS / QUESTIONS *** How do we select $r$ for deeply lazy maps when we don't have guidance from the structure of the posterior? This is not discussed. I find it interesting that the maps added at later iterations are the first ones applied to the source distribution (i.e. T = T_1 o ... o T_L, with samples from \rho going first through T_L, then T_{L-1}, etc.). Intuitively, I would think about the flow going in the other direction, where you would add maps to the end of a chain that gets closer and closer to the posterior (i.e. add T_L to T_{L-1} o ... o T_1 to get closer to \pi) as in something like Annealed Importance Sampling. However, this is not a criticsm, just an observation. I also find it appealing that the lazy framework provides benefits in both directions of the KL, but I wonder if there might be a disconnect between the bound you optimize (reverse KL) and the lazy construction (minimizing forward KL). In Figure 1, I think it would also be worthwhile to look at how a Gaussian distribution transport iteratively into the target banana distribution, rather than just the other way around. Table 1 notes: - Missing bracket on U-IAF variance diagnostic - Why is the G3-IAF trace metric bolded? All results there are within statistical error of each other. Figure 3: Why is there a jump in the diagnostic at $\ell = 9$? Also what is meant by "lack of precision", and how is this elucidated in Figure 3(b)? L273 typo: "due" -> "due to"


Review 2

Summary and Contributions: This paper proposes deep compositions of the lazy maps in order to adequately approximate the complex posterior. It also proves the weak convergence of the generated distributions to the posteriors. Finally, the experiments on several inference problems indicate that the benefits of the method: accuracy of inference, faster training, etc.

Strengths: The theoretical grounding is clear and sound. Four different problems have been explored with the method. It has clear explanations both in the main text and Appendix. Especially, the numerical algorithms are shown for the computation in the Appendix. The proposed seem a pretty good approach for flow-based methods.

Weaknesses: 1. The computation of H_i in deep lazy map could be time-consuming and with large variacne. It is not sure how to work well in real applications. 2. According to Proposition 3, we can see that there is a trade-off between coverage rate and optimal: when t = 1, the optimal do not alow faster converage. However, suboptimal choice without loss the convergence property.

Correctness: The claims and method are correct as I checked.

Clarity: It is a well-written paper.

Relation to Prior Work: Yes.

Reproducibility: Yes

Additional Feedback:


Review 3

Summary and Contributions: In this paper the authors introduce the idea of lazy maps, a novel Transport map approach to sample from high dimensional intractable distributions. The overall idea is to create a mapping from a simple base distribution such as a standard normal, such that the push through density approximates the posterior distribution of a target density of interest. The resulting algorithm consists of a number of steps including computation of an integral using importance sampling followed by eigen decomposition and finally in minimisation step to minimize KL Divergence between the pushforward distribution and a target posterior density of interest. The authors acknowledge that a single transformation may not suffice to approximate the target distribution interest and subsequently introduce deeply lazy maps where the final density is a composition of multiple individual lazy maps. they apply their approach to both an illustrative toy example based on the rotated banana distribution, Demonstrating repeat application of the pullback density results in successive Gaussianization of the resulting density. They consequently consider multiple real-world datasets, including Bayesian logistic regression on the UCI Parkinson’s classification dataset, and a Bayesian NN applied to the UCI yacht dataset.

Strengths: + Change of measure / change of variable flows from base distributions to induce complex distributions is an active area of research across multiple probabilistic ML domains + The inclusion of a diagnostic criterion that controls the expressiveness of the transformation by guaranteeing to bound the KL divergence between the true posterior and the transformed base distribution below a certain value for a certain computational budget, and the fact that this diagnostic is easily computable, is a particular strength

Weaknesses: One weakness seems to be the lack of comparison to established Bayesian inference methods: obviously the proposed method will lead to more expressive posteriors than e.g. mean field VI, but what is the associated cost, and in what regimes would one be preferable to the other? Similarly, given that the lazy map algorithm appears expensive (importance sampling, followed by an eigenvalue problem, followed by a KL divergence optimization), a comparison to even something like HMC would be useful to guide readers as to when the proposed framework is useful. The authors propose to use importance sampling to compute H but of course point out that the estimator can have high variance when \rho is far from \pi, and so propose to set the importance weights to one giving a biased but low variance estimator of H. It is not obvious why a biased low variance estimate is preferable to an unbiased high variance estimate. Similarly, one selling point of the overall approach is the ability to target high-dimensional spaces, but importance sampling may suffer here. While appendix section E deals with this, comparing tr(H) vs tr(H^B), can the authors show on a real example (possibly where the analytical posterior is calculable) what the effect of substituting H vs H^B is?

Correctness: The claims in the main paper appear to be correct: I have followed through what is written but not checked the supplementary proofs or previously referenced results in detail.

Clarity: Overall the paper is clear and well written. The one main comment is that the paper would benefit from separating out some of the prior work to make the original results stand out as novel, ie section 2 makes use of results from references [52, 25, 54], and as a reader not familiar with the previous references, it would be good if possible to separate these out as a prior work or prior results section, though it is understandable if this is infeasible. Line 88: if \pi_0 = \rho (prior) and the likelihood L = f(Ux), then the argument is that T \rho is the posterior \pi, but where is the guarantee this is a normalized probability distribution?

Relation to Prior Work: See comparison to established techniques comment under “Weaknesses”.

Reproducibility: Yes

Additional Feedback: Is there a reason the pdf isn’t rendered as text? It made selecting/highlighting not possible. Update: I have read the author's rebuttal, I had fairly minor queries and believe this remains a strong paper and will keep my score at 7


Review 4

Summary and Contributions: EDIT: I have read the authors' feedback and I maintain my original positive score. I look forward to seeing the community's response to this interesting and valuable work. --- This paper proposes a principled framework for learning a transport map/flow that renders tractable an otherwise intractable integral w.r.t. some posterior distribution. The transport map can have 1 or more layers. At layer l, the framework estimates the low-rank pushforward that is maximal with respect to the dimensions of greatest discrepancy (KL divergence) with the pullback of the posterior through the map at layer l-1. This greedy procedure can be bounded by a term proportional to the trace of a matrix H that looks similar to the fisher information but involves a ratio of the target and a standard normal density function. Since the goal is to transport the target to a standard normal, the largest eigenvalues of H at a particular layer l provide information on what gradient directions will minimize the remaining discrepancy. Estimating this matrix in order to perform eigendecomposition is the crux of the proposed methodology--one intractable integral is replaced with, hopefully, a more tractable one with some guarantees. This paper proposes a biased estimator which reduces variance but still (by the results of a previous paper) provides useful information towards Gaussianizing the target layer by layer.

Strengths: Theoretical grounding is perhaps the greatest strength of this paper. I enjoyed reading and was informed by the clear mathematical exposition, and appreciated the efforts by these authors to introduce tools that can certifiably capture low-dimensional structure as a means to do inference. However, this is the least novel part of the paper, as the majority of the theory has been worked out in prior works and is simply deployed here. The concept of a deeply lazy map and an algorithm for its construction is novel. The value of this contribution in terms of adding a new tool in low and moderate dimension cases is high.

Weaknesses: The scope of this methodology is not well explored, as I read it. The paper claims on line 68 that any inference problem is effectively decomposed into a series of low-dimensional problems, but it might be better to say that it is approximated thereby. It is unclear to me what problems would be best approximated this way from the experiments and analysis presented thus far. Similarly, the method is limited by being a greedy approach. This is not carefully discussed as far as I can tell. Consider the variable selection problem in generalized linear models, as an analogy. Greedy approaches that use residual information are known to yield suboptimal and incorrect regressions in many cases. Greedy suboptimality might explain why in section 4.4 the pullback sampling (and presumably the pullback distribution itself, as long as the MCMC was well tuned) fails to capture the lower-right square in the data generating field.

Correctness: I checked the proofs and am satisfied, although there is an unfortunate typo that renders line 441 in A.1 in error: the large paren and small paren have been switched, so that the determinant of the gradient of tau inverse at U transpose x is in the argument of the exponential. The empirical methodology is a little cautious, in that the problems tackled are relatively simple. Only a unimodal density is considered in the synthetic examples, e.g.

Clarity: I found the paper well written and a pleasure to read.

Relation to Prior Work: Yes, the work is well distinguished from previous contributions.

Reproducibility: Yes

Additional Feedback: I would like to see more discussion about different ways of reducing the variance of the estimator, other than introducing bias, e.g., control variates. Similarly, following on my comments above, I would like to see an experiment where this approach clearly fails. It should be possible to design a synthetic problem where a certain rank approximation for each unitary leads the estimator astray. There's no free lunch of course and it would valuable to know when I should expect to pay for one when I use this framework.

[Author Response · NeurIPS 2020]

We are very grateful to all the reviewers for their thoughtful feedback. Below we address the main questions and comments, and identify topics we will expand on in the final paper. All typos and minor points will also be fixed.

**Theoretical questions on Prop. 3.** The sequence of distributions that converges weakly to $\pi$ is $\{(\mathfrak{T}_\ell)_\sharp \rho\}_{\ell \geq 1}$ as the number of lazy layers increases, i.e., $\ell \to \infty$. Crucially, condition (6) must apply simultaneously to *all* layers for a given $0 < t \leq 1$, rather than one particular layer. In the future, we plan to generalize our result by allowing $t \equiv (t_\ell)_{\ell \geq 0}$ to be a sequence that goes to zero sufficiently slowly to maintain the weak convergence of $\{(\mathfrak{T}_\ell)_\sharp \rho\}_{\ell \geq 1}$. In the case of $r = 1$, each layer operates on a one-dimensional subspace, but each subspace can be different, and thus we can capture the posterior in the limit. We do not claim any specific rate of convergence for the approximate posteriors based on $t$, though it is reasonable to suspect that $t$ close to 1 will yield faster convergence. We are currently considering methods for estimating $t$ in future work, which would lead to empirical studies of the convergence rate and $t$.

**Greedy sub-optimality.** Prop. 3 implies that any inference problem can be decomposed into a sequence of $r$-dimensional problems, and that the limit of this sequence is exactly the posterior. Therefore, while our greedy method is certainly not optimal for a given length $\ell$, we can't provide an example where a certain rank $r$ leads the approximate posterior astray in the limit $\ell \to \infty$. However, deeply lazy maps can sometimes suffer from a certain greedy sub-optimality with regard to the rank and the choice of transport class. For example, the target distribution in the toy problem of §4.1 can be captured with a single full-dimensional quadratic map. By choosing $r = 1$, we instead need a sequence of maps to capture the target. This is an example where the underlying problem does not have immediate lazy structure, and applying an overly lazy map requires extra work with no pay-off. This naturally raises the question of how to choose $r$. The spectrum of $H$ is a natural guide: if it decays quickly, then one can fix $r$ using an error threshold as discussed in Prop. 2. Otherwise, the choice of $r$ defines a trade-off between how expensive each layer is to train and how many layers one needs to train. One strategy is to limit $r$ by some $r_{\max}$, which defines a maximum computational budget for each layer. Another consideration, as highlighted by the example of §4.3, is that reducing the dimension of each layer can *improve* training behavior. The missing square in the posterior realizations of the diffusion field in §4.4 is in fact a property of the true posterior; we have verified this fact with MCMC results.

**Comparison to MCMC.** Finding a fair comparison between MCMC and VI is a universal and interesting question in Bayesian computation, as the two methods have different computational cost patterns. In VI one spends considerable computational effort to construct the approximate posterior, but afterwards has cheap access to (approximate) samples and normalized density evaluations. How well the approximation matches the true posterior depends on the expressiveness of the transport map/flow and the ability to optimize the map—two qualities that the lazy map framework seems to improve. MCMC methods in contrast require continual computational effort (even after tuning), but (asymptotically) generate samples from the exact posterior. We will discuss these trade-offs to additionally frame our contribution in the final paper. The strongest comparison we believe we can make is comparing the ESS of an MCMC method with and without transport preconditioning (as discussed in [23, 41] and in §4.4). We will make these improvements a more central measure of success and will also report the improvements when applied to the examples of §4.2 and §4.3.

**Computing $H$ considerations.** The cost of forming $H$ scales with the cost of computing the gradient of the un-normalized target log-density at a sample. This is required for each optimization step as well. For the example of §4.2, the cost of computing $H$ with 500 samples is the same as that of 5 optimization steps with 100 samples each, a relatively minor cost compared to 20000 total optimization steps. The cost of identifying the dominant eigenspace is also small in comparison. In the final paper we will include this cost in the training plots in Appendix G, either by plotting against wall clock time or the number of gradient evaluations. We do accept that the cost of map evaluations increases as the lazy map grows deeper, which also occurs when increasing the length of a typical normalizing flow. Empirically, we have found that the dominant subspace $U_r$ of $H^B$ does not differ strongly from that of $H$, but that the smaller variance of the estimator $\widehat{H}^B$ can yield more reliable results in early iterations. Methods to reduce the variance of an unbiased estimator of $H$, such as control variates, may certainly be useful in other problems, and we will provide more guidance on this in the final paper. We will also discuss a criterion for switching to the importance sampling estimator of $H$ for deeper layers. As the intermediate targets become closer to Gaussian, the variance of this unbiased estimator naturally reduces. Currently, however, we haven't found problems where the basis derived from $H^B$ has been ineffective.

**Other points.** In the discussion after Prop. 1, we should have $\mathcal{L}_y(x) \propto f(U^T x)$, i.e. $f$ is implicitly scaled to enforce normalization. The phrase "lack of precision" in §4.4 refers to the finite number of samples drawn from $\rho$ used to resolve the objective and diagnostics. We believe the jump in the diagnostic at $\ell = 9$ is the result of sampling noise when computing $H_9$, as the diagnostic drops back after one additional lazy step. We agree that the ordering of our map composition is the reverse of that in several methods. This is a result of defining the residual targets $\pi_\ell$ as pullbacks, and allows all maps trained to use the Gaussian base distribution. Outside of this work, we are considering the effects of using the forward KL in the error bound and the backward KL in training, though we observe benefits in both directions. We agree with several reviewers that the addition of a dedicated prior work section will improve the organization of the paper, and we will include additional derivations of key results in the appendices to make things self-contained.

[Meta-Review · NeurIPS 2020]

A nice work in an important research area.